# A century of trends in adult human height

## NCD Risk Factor Collaboration (NCD-RisC)*

**Abstract** Being taller is associated with enhanced longevity, and higher education and earnings. We reanalysed 1472 population-based studies, with measurement of height on more than 18.6 million participants to estimate mean height for people born between 1896 and 1996 in 200 countries. The largest gain in adult height over the past century has occurred in South Korean women and Iranian men, who became 20.2 cm (95% credible interval 17.5–22.7) and 16.5 cm (13.3–19.7) taller, respectively. In contrast, there was little change in adult height in some sub-Saharan African countries and in South Asia over the century of analysis. The tallest people over these 100 years are men born in the Netherlands in the last quarter of 20th century, whose average heights surpassed 182.5 cm, and the shortest were women born in Guatemala in 1896 (140.3 cm; 135.8–144.8). The height differential between the tallest and shortest populations was 19-20 cm a century ago, and has remained the same for women and increased for men a century later despite substantial changes in the ranking of countries.

## Introduction

Being taller is associated with enhanced longevity, lower risk of adverse pregnancy outcomes and cardiovascular and respiratory diseases, and higher risk of some cancers (*Paajanen et al., 2010*; *Emerging Risk Factors Collaboration, 2012*; *Green et al., 2011*; *Nelson et al., 2015*; *Batty et al., 2010*; *World Cancer Research Fund / American Institute for Cancer Research, 2007*; *2010*; *2011*; *2012*; *2014a*; *2014b*; *Nüesch et al., 2015*; *Davies et al., 2015*; *Zhang et al., 2015*; *Kozuki et al., 2015*; *Black et al., 2008*). There is also evidence that taller people on average have higher education, earnings, and possibly even social position (*Adair et al., 2013*; *Stulp et al., 2015*; *Barker et al., 2005*; *Strauss and Thomas, 1998*; *Chen and Zhou, 2007*; *Case and Paxson, 2008*).

Although height is one of the most heritable human traits (*Fisher, 1919*; *Lettre, 2011*), cross-population differences are believed to be related to non-genetic, environmental factors. Of these, foetal growth (itself related to maternal size, nutrition and environmental exposures), and nutrition and infections during childhood and adolescence are particularly important determinants of height during adulthood (*Cole, 2000*; *Silventoinen et al., 2000*; *Dubois et al., 2012*; *Haeffner et al., 2002*; *Sørensen et al., 1999*; *Victora et al., 2008*; *Eveleth and Tanner, 1990*; *Tanner, 1962*; *Tanner, 1992*; *Bogin, 2013*). Information on height, and its trends, can therefore help understand the health impacts of childhood and adolescent nutrition and environment, and of their social, economic, and political determinants, on both non-communicable diseases (NCDs) and on neonatal health and survival in the next generation (*Cole, 2000*; *Tanner, 1992*; *Tanner, 1987*).

Trends in men's height have been analysed in Europe, the USA, and Japan for up to 250 years, using data on conscripts, voluntary military personnel, convicts, or slaves (*Cole, 2000*; *Floud et al., 1990*; *Fogel et al., 1983*; *Schmidt et al., 1995*; *Floud et al., 2011*; *Tanner et al., 1982*; *Hatton and Bray, 2010*; *Tanner, 1981*; *Facchini and Gualdi-Russo, 1982*). There are fewer historical data for women, and for other regions where focus has largely been on children and where adult data tend to be reported at one point in time or over short periods (*Subramanian et al., 2011*; *Grasgruber et al., 2014*; *Baten and Blum, 2012*; *Deaton, 2007*; *Mamidi et al., 2011*; *van Zanden et al., 2014*). In this paper, we pooled worldwide population-based data to estimate height in adulthood for men and women born over a whole century throughout the world.

*For correspondence: majid. ezzati@imperial.ac.uk

**Competing interests:** The authors declare that no competing interests exist.

**eLife digest** People from different countries grow to different heights. This may be partly due to genetics, but most differences in height between countries have other causes. For example, children and adolescents who are malnourished, or who suffer from serious diseases, will generally be shorter as adults. This is important because taller people generally live longer, are less likely to suffer from heart disease and stroke, and taller women and their children are less likely to have complications during and after birth. Taller people may also earn more and be more successful at school. However, they are also more likely to develop some cancers.

The NCD Risk Factor Collaboration set out to find out how tall people are, on average, in every country in the world at the moment, and how this has changed over the past 100 years. The analysis revealed large differences in height between countries. The tallest men were born in the last part of the 20th century in the Netherlands, and were nearly 183 cm tall on average. The shortest women were born in 1896 in Guatemala, and were on average 140 cm tall. The difference between the shortest and tallest countries is about 20 cm for both men and women. This means there are large differences between countries in terms of nutrition and the risk of developing some diseases.

The way in which height has changed over the past 100 years also varies from country to country. Iranian men born in 1996 were around 17 cm taller than those born in 1896, and South Korean women were 20 cm taller. In other parts of the world, particularly in South Asia and parts of Africa, people are only slightly taller than 100 years ago, and in some countries people are shorter than they were 50 years ago.

There is a need to better understand why height has changed in different countries by different amounts, and use this information to improve nutrition and health across the world. It would also be valuable to understand how much becoming taller has been responsible for improved health and longevity throughout the world.

## Results

We estimated that people born in 1896 were shortest in Asia and in Central and Andean Latin America (*Figure 1* and *Figure 2*). The 1896 male birth cohort on average measured only 152.9 cm (credible interval 147.9–157.9) in Laos, which is the same as a well-nourished 12.5-year boy according to international growth standards (*de Onis et al., 2007*), followed by Timor-Leste and Guatemala. Women born in the same year in Guatemala were on average 140.3 cm (135.8–144.8), the same as a well-nourished 10-year girl. El Salvador, Peru, Bangladesh, South Korea and Japan had the next shortest women. The tallest populations a century ago lived in Central and Northern Europe, North America and some Pacific islands. The height of men born in Sweden, Norway and the USA surpassed 171 cm, ~18–19 cm taller than men in Laos. Swedish women, with average adult height of 160.3 cm (158.2–162.4), were the tallest a century ago and 20 cm taller than women in Guatemala. Women were also taller than 158 cm in Norway, Iceland, the USA and American Samoa.

Changes in adult height over the century of analysis varied drastically across countries. Notably, although the large increases in European men's heights in the 19th and 20th century have been highlighted, we found that the largest gains since the 1896 birth cohort occurred in South Korean women and Iranian men, who became 20.2 cm (17.5–22.7) and 16.5 cm (13.3–19.7) taller, respectively (*Figure 3*, *Figure 4* and *Figure 5*). As a result, South Korean women moved from the fifth shortest to the top tertile of tallest women in the world over the course of a century. Men in South Korea also had large gains relative to other countries, by 15.2 cm (12.3–18.1). There were also large gains in height in Japan, Greenland, some countries in Southern Europe (e.g., Greece) and Central Europe (e.g., Serbia and Poland, and for women Czech Republic). In contrast, there was little gain in height in many countries in sub-Saharan Africa and South Asia.

The pace of growth in height has not been uniform over the past century. The impressive rise in height in Japan stopped in people born after the early 1960s (*Figure 6*). In South Korea, the flattening began in the cohorts born in the 1980s for men and it may have just begun in women. As a result, South Korean men and women are now taller than their Japanese counterparts. The rise is

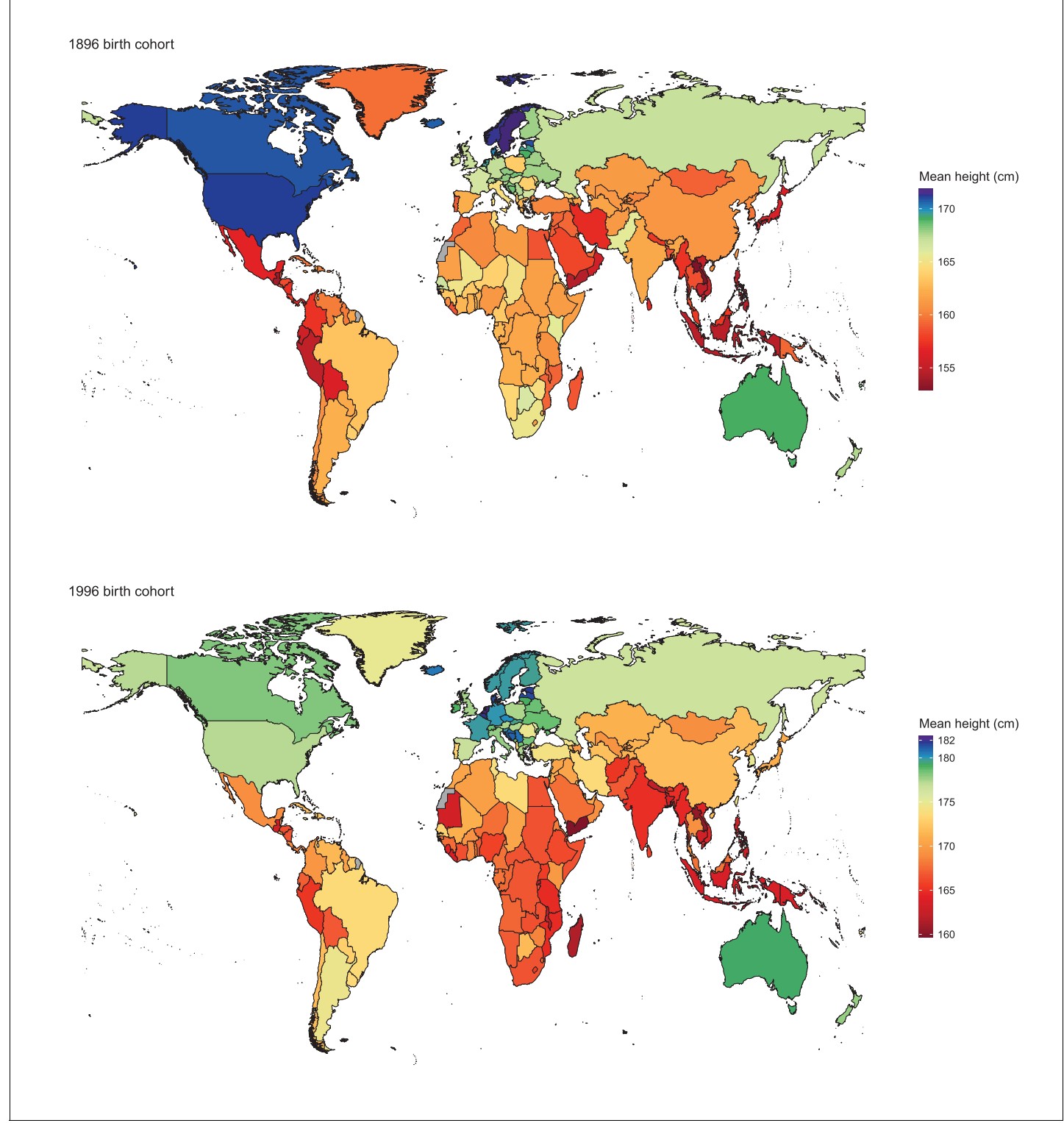

**Figure 1.** Adult height for the 1896 and 1996 birth cohorts for men. See www.ncdrisc.org for interactive version.

continuing in other East and Southeast Asian countries like China and Thailand, with Chinese men and women having surpassed the Japanese (but not yet as tall as South Koreans). The rise in adult

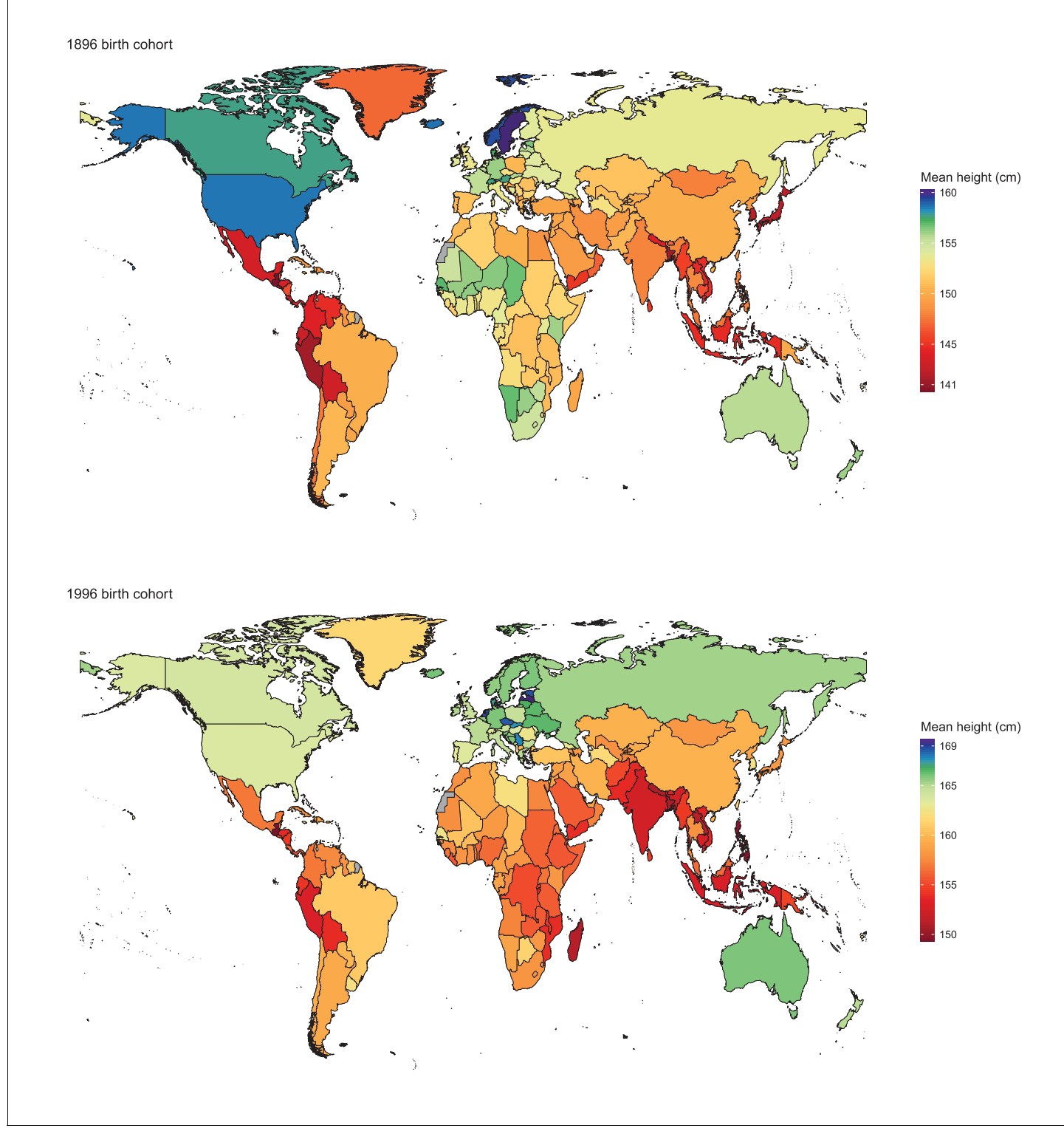

**Figure 2.** Adult height for the 1896 and 1996 birth cohorts for women. See www.ncdrisc.org for interactive version.

height also seems to have plateaued in South Asian countries like Bangladesh and India at much lower levels than in East Asia, e.g., 5–10 cm shorter than it did in Japan and South Korea.

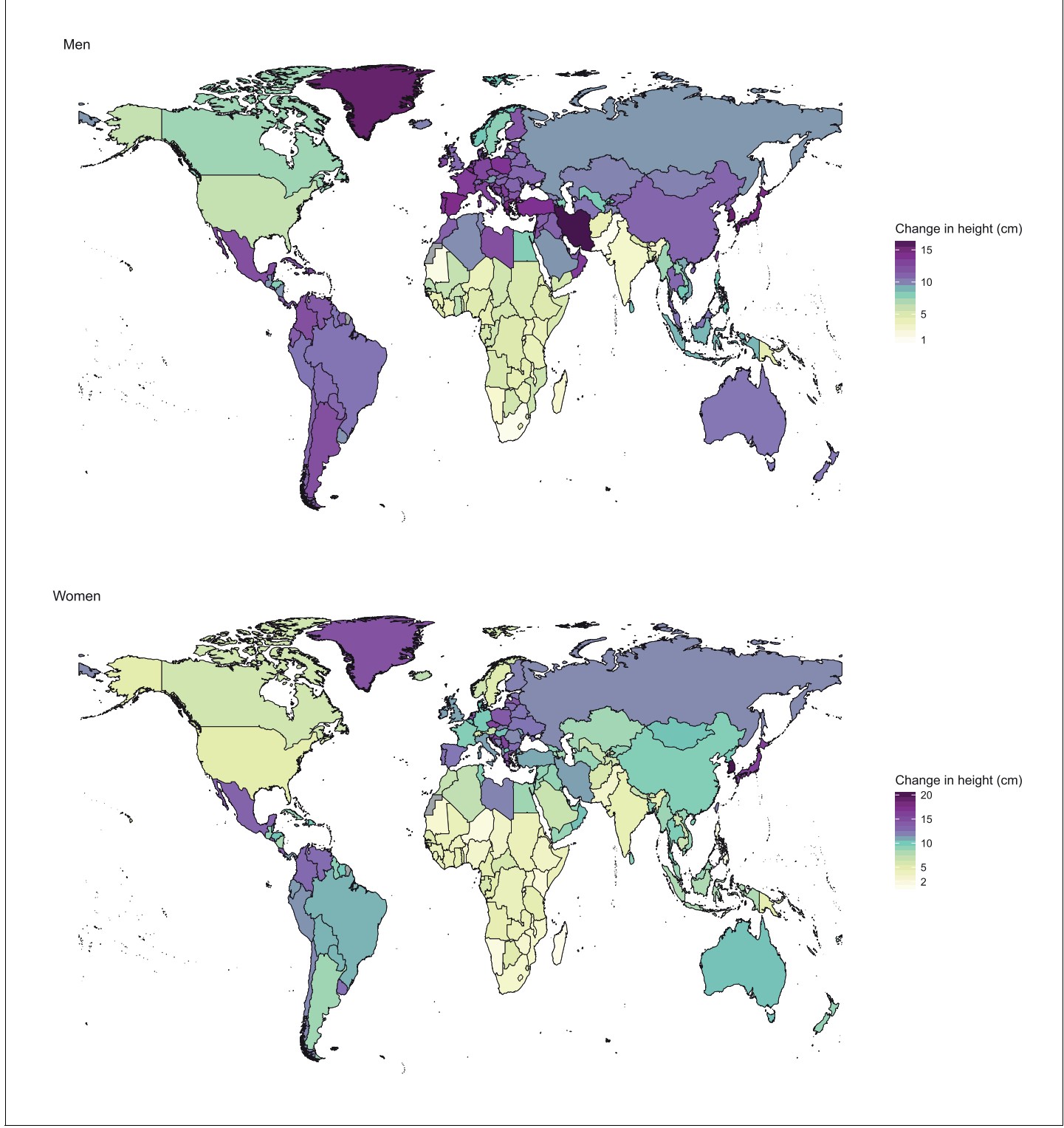

**Figure 3.** Change in adult height between the 1896 and 1996 birth cohorts.

There were also variations in the time course of height change across high-income western countries, with height increase having plateaued in Northern European countries like Finland and in English-speaking countries like the UK for 2–3 decades (*Larnkaer et al., 2006*; *Schönbeck et al.,*

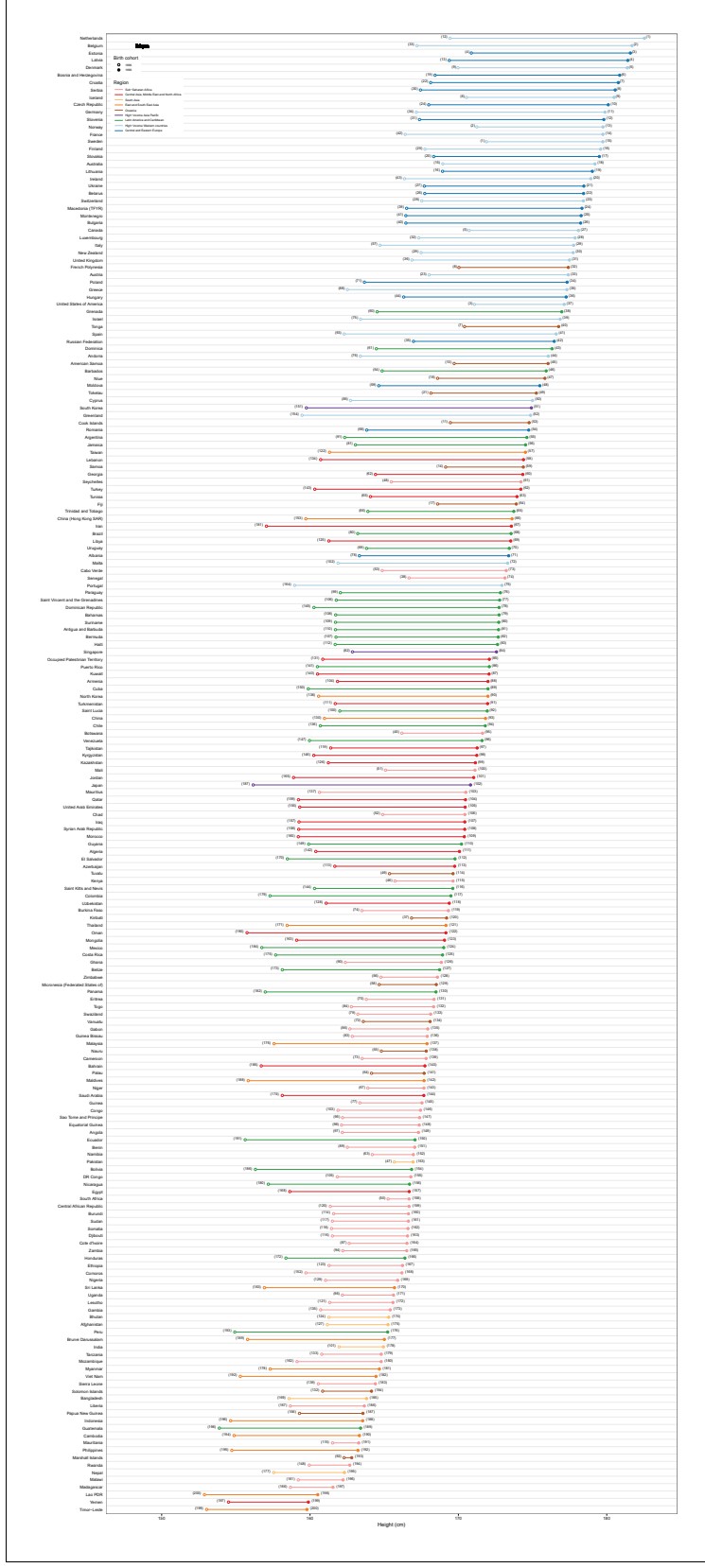

**Figure 4.** Height in adulthood for the 1896 and 1996 birth cohorts for men. The open circle shows the adult height attained by the 1896 birth cohort and the filled circle that of the 1996 birth cohort; the length of the connecting
*Figure 4 continued on next page*

*Figure 4 continued*
line represents the change in height over the century of analysis. The numbers next to each circle show the country's rank in terms of adult height for the corresponding cohort. See www.ncdrisc.org for interactive version.

*2013*), followed by Eastern Europe (*Figure 7*). The earliest of these occurred in the USA, which was one of the tallest nations a century ago but has now fallen behind its European counterparts after having had the smallest gain in height of any high-income country (*Tanner, 1981*; *Komlos and Lauderdale, 2007*; *Komlos and Baur, 2004*; *Sokoloff and Villaflor, 1982*). In contrast, height is still increasing in some Southern European countries (e.g., Spain), and in many countries in Latin America.

As an exception to the steady gains in most countries, adult height decreased or at best remained the same in many countries in sub-Saharan Africa for cohorts born after the early 1960s, by around 5 cm from its peak in some countries (see for example Niger, Rwanda, Sierra Leone, and Uganda in *Figure 8*). More recently, the same seems to have happened for men, but not women, in some countries in Central Asia (e.g., Azerbaijan and Uzbekistan) and Middle East and North Africa (e.g., Egypt and Yemen), whereas in others (e.g., Iran) both sexes continue to grow taller.

Men born in 1996 surpass average heights of 181 cm in the Netherlands, Belgium, Estonia, Latvia and Denmark, with Dutch men, at 182.5 cm (180.6–184.5), the tallest people on the planet. The gap with the shortest countries – Timor-Leste, Yemen and Laos, where men are only ~160 cm tall – is 22–23 cm, an increase of ~4 cm on the global gap in the 1896 birth cohort. Australia was the only non-European country where men born in 1996 were among the 25 tallest in the world. Women born in 1996 are shortest in Guatemala, with an average height of 149.4 cm (148.0–150.8), and are shorter than 151 cm in the Philippines, Bangladesh and Nepal. The tallest women live in Latvia, the Netherlands, Estonia and Czech Republic, with average height surpassing 168 cm, creating a 20 cm global gap in women's height (*Figure 5*).

Male and female heights were correlated across countries in 1896 as well as in 1996. Men were taller than women in every country, on average by ~11 cm in the 1896 birth cohort and ~12 cm in the 1996 birth cohort (*Figure 9*). In the 1896 birth cohort, the male-female height gap in countries where average height was low was slightly larger than in taller nations. In other words, at the turn of the 20th century, men seem to have had a relative advantage over women in undernourished compared to better-nourished populations. A century later, the male-female height gap is about the same throughout the height range. Changes in male and female heights over the century of analysis were also correlated, which is in contrast to low correlation between changes in male and female BMIs as reported elsewhere (*NCD Risk Factor Collaboration, 2016*).

Change in population mean height was not correlated with change in mean BMI (*NCD Risk Factor Collaboration, 2016*) across countries for men (correlation coefficient = −0.016) and was weakly inversely correlated for women (correlation coefficient = −0.28) (*Figure 10*). Countries like Japan, Singapore and France had larger-than-median gains in height but little change in BMI, in contrast to places like the USA and Kiribati where height has increased less than the worldwide median while BMI has increased a great deal.

## Discussion

We found that over the past century adult height has changed substantially and unevenly in the world's countries, with no indication of convergence across countries. The height differential between the tallest and shortest populations was ~19 cm for men and ~20 cm for women a century ago, and has remained about the same for women and increased for men a century later despite substantial changes in the ranking of countries in terms of adult height.

Data from military conscripts and personnel have allowed reconstructing long-term trends in height in some European countries and the USA, albeit largely for men, and treating it as a 'mirror' to social and environmental conditions that affect nutrition, health and economic prosperity, in each generation and across generations (*Tanner, 1987*; *Fogel, 2004*; *Komlos, 2009*; *Martins et al., 2014*; *Martorell, 1995*). Our results on the large gains in continental European countries, and that they have overtaken English-speaking countries like the USA, are consistent with these earlier studies

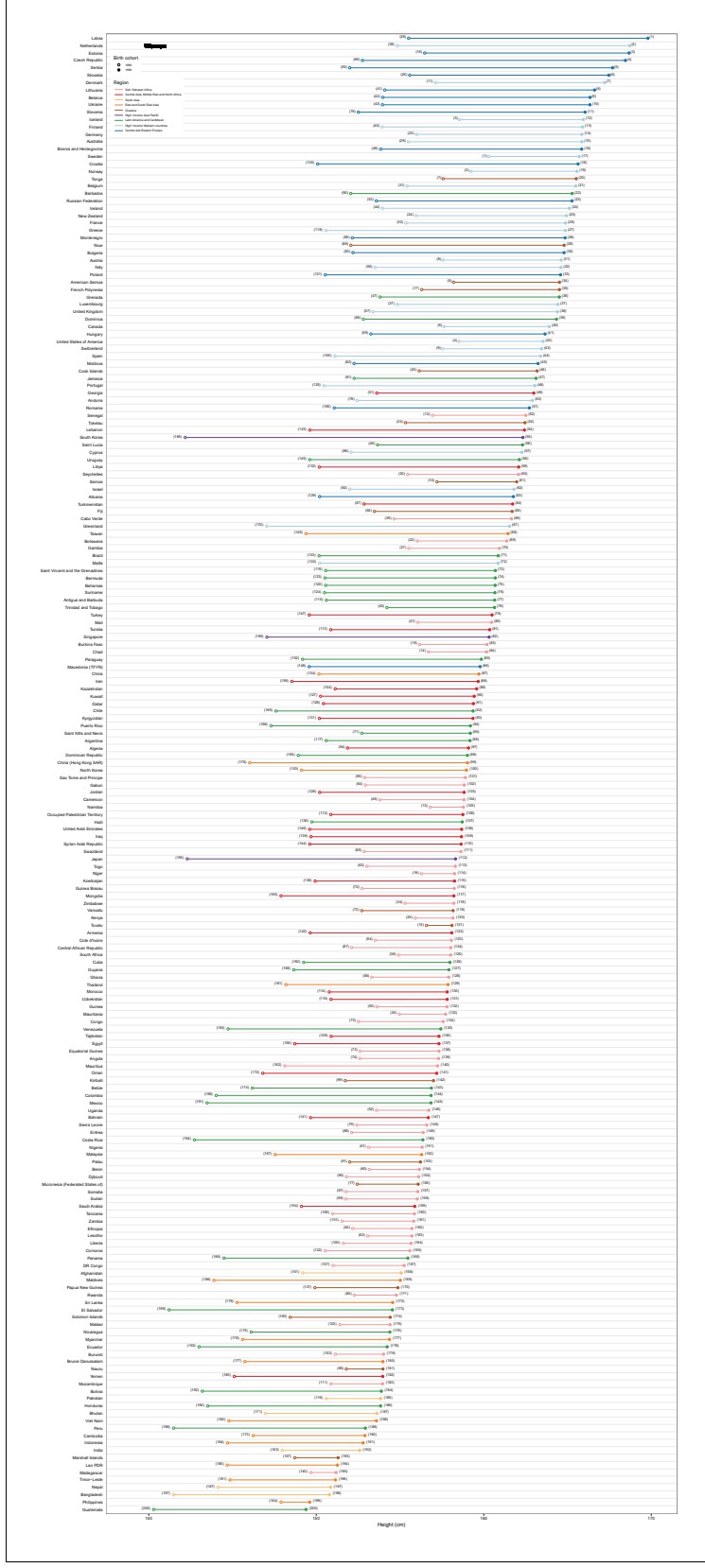

**Figure 5.** Height in adulthood for the 1896 and 1996 birth cohorts for women. The open circle shows the adult height attained by the 1896 birth cohort and the filled circle that of the 1996 birth cohort; the length of the
*Figure 5 continued on next page*

*Figure 5 continued*

connecting line represents the change in height over the century of analysis. The numbers next to each circle show the country's rank in terms of adult height for the corresponding cohort. See www.ncdrisc.org for interactive version.

although these earlier analyses covered fewer countries in Eastern and Southern Europe, and used some self-reported data with simple adjustments that cannot fully correct for their bias (*Hatton and Bray, 2010*; *Facchini and Gualdi-Russo, 1982*; *Baten and Blum, 2012*).

Less has been known about trends in women's height, and those in non-English-speaking/non-European parts of the world. We found that some of the most important changes in height have happened in these under-investigated populations. In particular, South Korean and Japanese men and women, and Iranian men, have had larger gains than European men, and similar trends are now happening in China and Thailand. These gains may partially account for the fact that women in Japan and South Korea have achieved the first and fourth highest life expectancy in the world (see also below). In contrast to East Asia's impressive gains, the rise in height seems to have stopped early in South Asia and reversed in Africa, reversing or diminishing Africa's earlier advantage over Asia. Prior studies have documented a rise in stunting in children in sub-Saharan Africa which continued to the

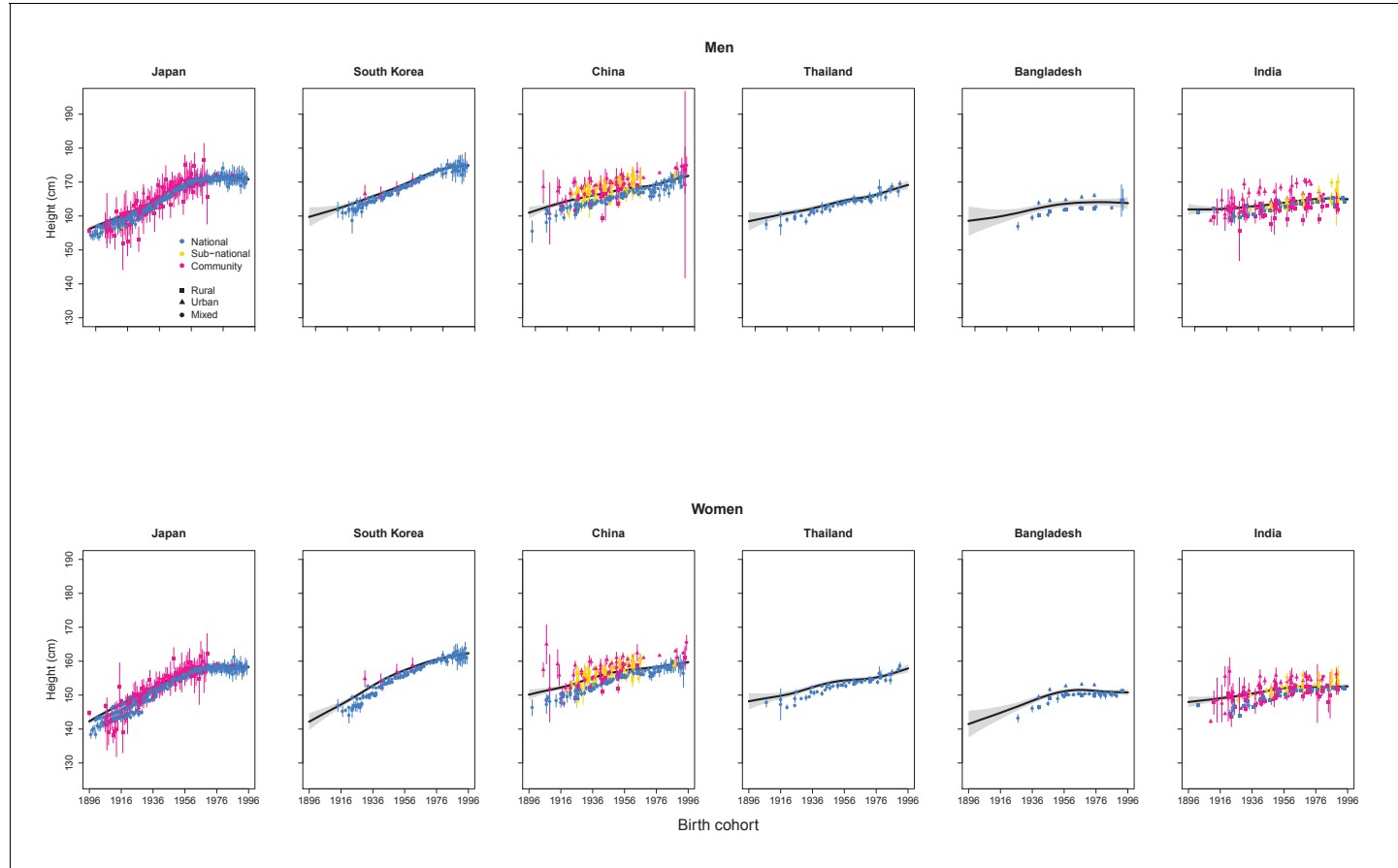

**Figure 6.** Trends in height for the adult populations of selected countries in Asia. The solid line represents the posterior mean and the shaded area the 95% credible interval of the estimates. The points show the actual data from each country, together with its 95% confidence interval due to sampling. The solid line and shaded area show estimated height at 18 years of age, while the data points show height at the actual age of measurement. The divergence between estimates and data for earlier birth cohorts is because participants from these birth cohorts were generally older when their heights were measured.

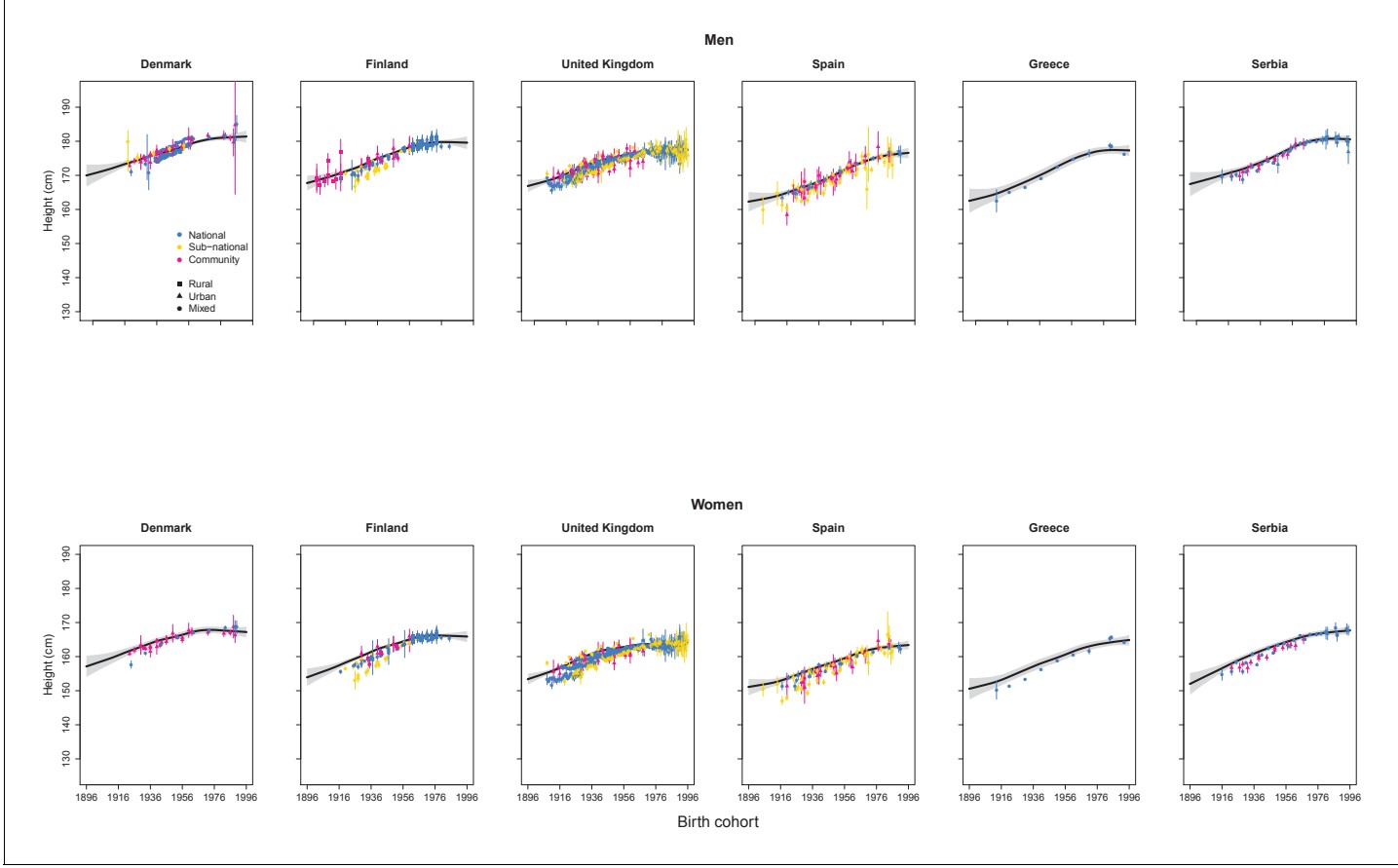

**Figure 7.** Trends in height for the adult populations of selected countries in Europe. The solid line represents the posterior mean and the shaded area the 95% credible interval of the estimates. The points show the actual data from each country, together with its 95% confidence interval due to sampling. The solid line and shaded area show estimated height at 18 years of age, while the data points show height at the actual age of measurement. The divergence between estimates and data for earlier birth cohorts is because participants from these birth cohorts were generally older when their heights were measured.

mid-1990s (*Stevens et al., 2012*). Our results indicate that such childhood adversity may have carried forward to adulthood and be affecting health in the region. The early African advantage over Asia may also have been partly due to having a more diverse diet compared to the vegetable and cereal diet in Asia, partly facilitated by lower population density (*Deaton, 2007*; *Moradi, 2010*). Rising population, coupled with worsening economic status during structural adjustment, may have undermined earlier dietary advantage (*Stevens et al., 2012*; *Pongou et al., 2006*; *Weil et al., 1990*; *Sundberg, 2009*).

The main strengths of our study are its novel scope of estimating a century of trends in adult height for all countries in the world and for both sexes. Our population-based results complement the individual-level studies on the genetic and environmental determinants of within-population variation in height, and will help develop and test hypotheses about the determinants of adult height, and its health consequences. We achieved this by using a large number of population-based data sources from all regions of the world. We put particular emphasis on data quality and used only population-based data that had measured height, which avoids bias in self-reported height. Data were analysed according to a common protocol before being pooled, and characteristics and quality of data sources were verified through repeated checks by Collaborating Group members. Finally, we pooled data using a statistical model that could characterize non-linear trends and that used all available data while giving more weight to national data than to subnational and community surveys.

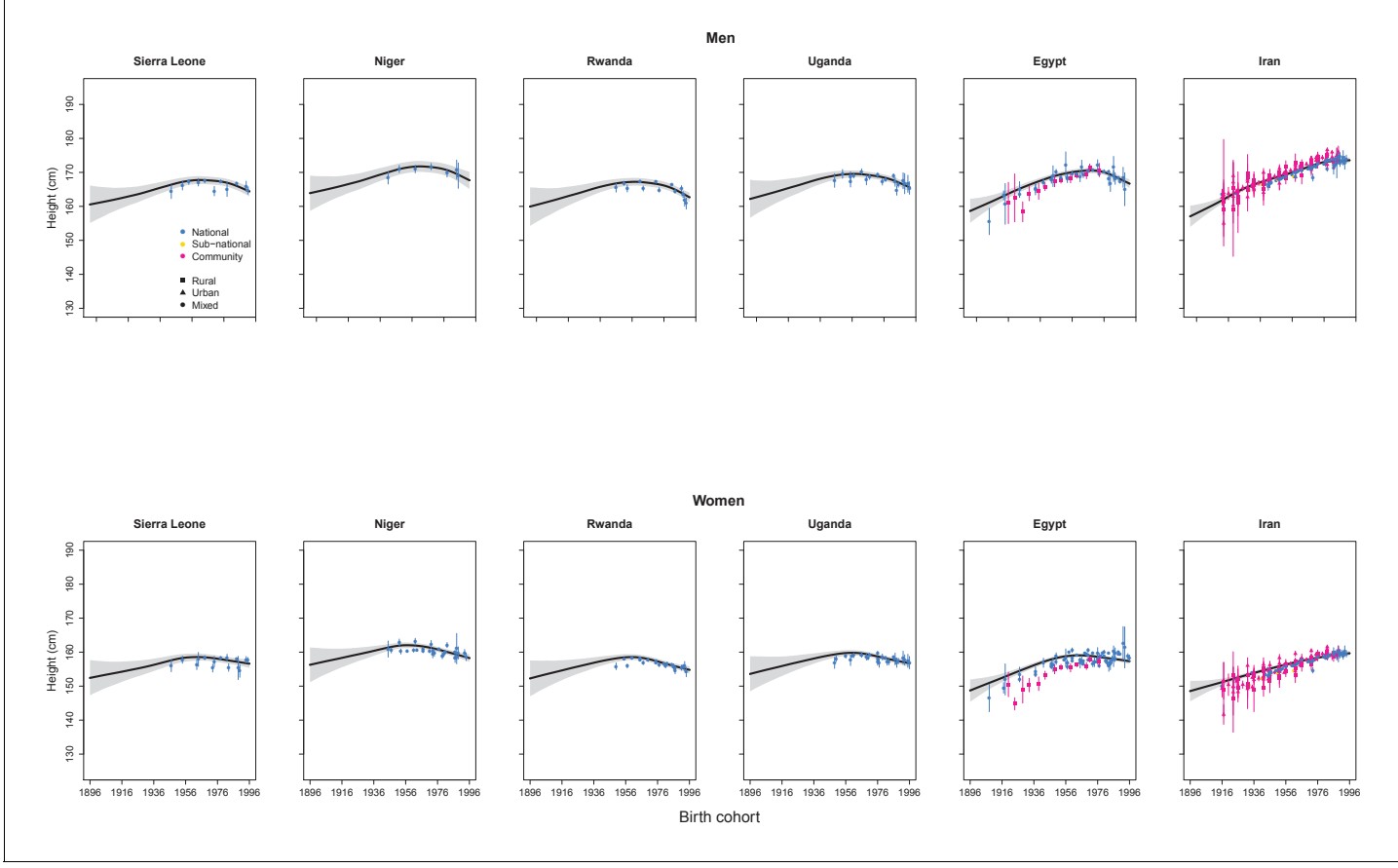

**Figure 8.** Trends in height for the adult populations of selected countries in the Middle East, North Africa, and sub-Saharan Africa. The solid line represents the posterior mean and the shaded area the 95% credible interval of the estimates. The points show the actual data from each country, together with its 95% confidence interval due to sampling. The solid line and shaded area show estimated height at 18 years of age, while the data points show height at the actual age of measurement. The divergence between estimates and data for earlier birth cohorts is because participants from these birth cohorts were generally older when their heights were measured.

Although we have gathered an unprecedentedly comprehensive database of human height and growth, and have applied a statistical model that maximally utilizes the information in these sources, data in some countries were rather limited or were from community or sub-national studies. This is reflected in larger uncertainty of the estimated height in these countries. To overcome this, surveillance of growth, which has focused largely on children, should also systematically monitor adolescents and adults given the increasingly abundant evidence on their effects on adult health and human capital. Even measured height data can be subject to measurement error depending on how closely study protocols are followed. Finally, we did not have separate data on leg and trunk lengths, which may differ in their determinants, especially in relation to age at menarche and pre- vs. post-pubertal growth and nutrition, and health effects (*Tanner et al., 1982*; *Frisch and Revelle, 1971*).

Greater height in adulthood is both beneficially (cardiovascular and respiratory diseases) and harmfully (colorectal, postmenopausal breast and ovarian cancers, and possibly pancreatic, prostate and premenopausal breast cancers) associated with several diseases, independently of its inverse correlation with BMI (*Paajanen et al., 2010*; *Emerging Risk Factors Collaboration, 2012*; *Green et al., 2011*; *Nelson et al., 2015*; *Batty et al., 2010*; *World Cancer Research Fund / American Institute for Cancer Research, 2007*; *2010*; *2011*; *2012*; *2014a*; *2014b*; *Nüesch et al., 2015*; *Davies et al., 2015*; *Zhang et al., 2015*). If the associations in epidemiological studies are causal, which is supported by the more recent evidence from Mendelian randomisation studies (*Green et al., 2011*; *Nüesch et al., 2015*; *Davies et al., 2015*; *Zhang et al., 2015*), the ~20 cm height range in the world is associated with a 17% lower risk of cardiovascular mortality and 20–40%

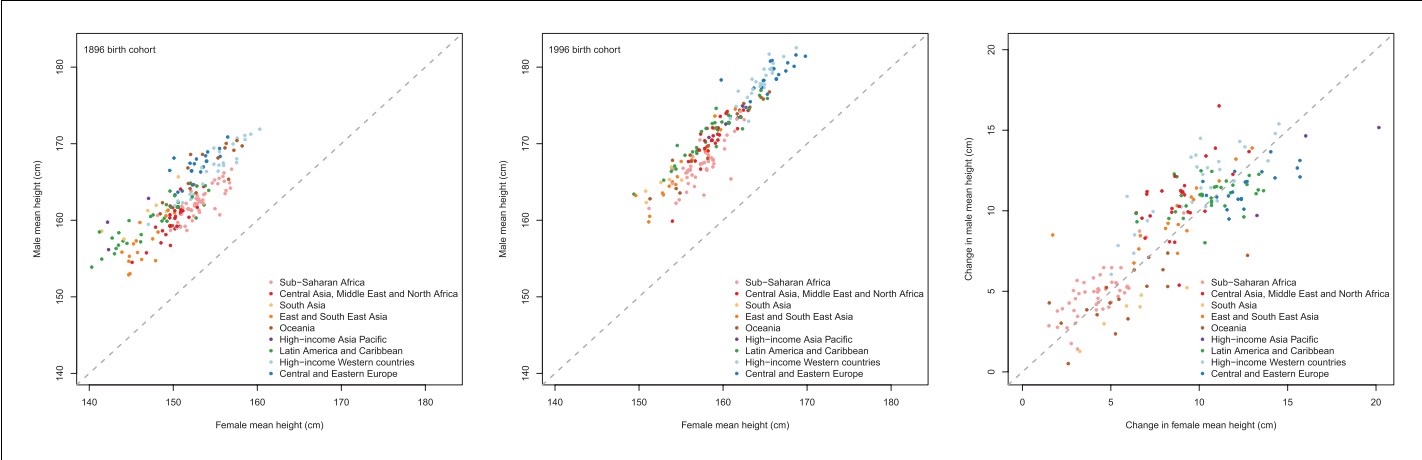

**Figure 9.** Height in adulthood for men vs. women for the 1896 and 1996 birth cohorts, and change in men's vs. women's heights from 1896 to 1996.

higher risk of various site-specific cancers, in tall versus short countries. Consistent with individual-level evidence on the association between taller height and lower all-cause mortality in adult ages (*Emerging Risk Factors Collaboration, 2012*), gains in mean population height in successive cohorts are associated with lower mortality in middle and older ages in countries with reliable mortality data (correlation coefficient = −0.58 for men and −0.68 for women) (*Figure 11*), demonstrating the large impacts of height gain on population health and longevity. Further, short maternal stature increases the risk of small-for-gestational-age and preterm births, both risk factors for neonatal mortality, and of pregnancy complications (*Kozuki et al., 2015*; *Black et al., 2008*). Therefore, improvements vs. stagnation in women's height can influence trends in infant and maternal mortality.

Our study also shows the potential for using height in early adulthood as an indicator that integrates across different dimensions of sustainable human development. Adult height signifies not only foetal and early childhood nutrition, which was included in the Millennium Development Goals, but also that of adolescents (*Lancet, 2014*). Further, adult height is a link between these early-life experiences and NCDs, longevity, education and earnings. It can easily be measured in health surveys and can be used to investigate differences across countries and trends over time, as done in our work, as well as within-country inequalities. Therefore, height in early adulthood, which varies substantially across countries and over time, provides a measurable indicator for sustainable development, with links to health and longevity, nutrition, education and economic productivity.

## Materials and methods

### Overview

We estimated trends in mean height for adults born from 1896 to 1996 (i.e., people who had reached their 18th birthday from 1914 to 2014) in 200 countries and territories. Countries were organized into 20 regions, mostly based on a combination of geography and national income (*Supplementary file 1*). Our study had two steps, described below. First, we identified, accessed, and re-analysed population-based measurement studies of human anthropometry. We then used a statistical model to estimate trends for all countries and territories.

### Data sources

We used data sources that were representative of a national, subnational, or community population and had measured height. We did not use self-reported height because it is subject to systematic bias that varies by geography, time, age, sex, and socioeconomic characteristics like education and ethnicity (*Engstrom et al., 2003*; *Connor Gorber et al., 2007*; *Wetmore and Mokdad, 2012*; *Schenker et al., 2010*; *Ezzati et al., 2006*; *Clarke et al., 2014*; *Hayes et al., 2011*).

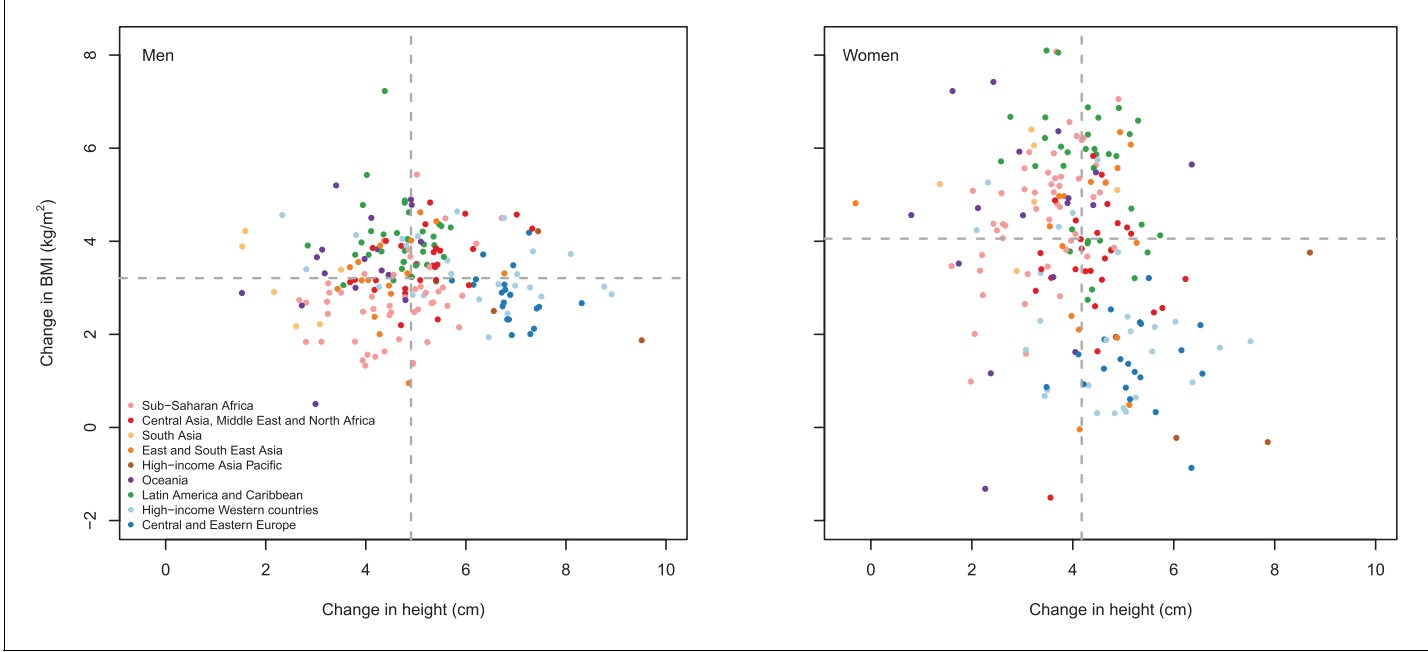

**Figure 10.** Change, over the 1928–1967 birth cohorts, in mean BMI vs. in mean height. Each point shows one country. BMI change was calculated for mean BMI at 45–49 years of age – an age when diseases associated with excess weight become common but weight loss due to pre-existing disease is still uncommon. BMI data were available for 1975–2014 (***NCD Risk Factor Collaboration, 2016***); 45–49 year olds in these years correspond to 1928–1967 birth cohorts. BMI data were from a pooled analysis of 1698 population-based measurement studies with 19.2 million participants, with details reported elsewhere (***NCD Risk Factor Collaboration, 2016***).

Data sources were included in the NCD-RisC database if:

- measured data on height, weight, waist circumference, or hip circumference were available;
- study participants were five years of age and older;
- data were collected using a probabilistic sampling method with a defined sampling frame;
- data were representative of the general population at the national, subnational, or community level;
- data were from the countries and territories listed in ***Supplementary file 1***.

We excluded data sources on population subgroups whose anthropometric status may differ systematically from the general population, including:

- studies that had included or excluded people based on their health status or cardiovascular risk;
- ethnic minorities;
- specific educational, occupational, or socioeconomic subgroups of the population; and
- those recruited through health facilities, with the exception noted below.

We used school-based data in countries where secondary school enrolment was 70% or higher, and used data whose sampling frame was health insurance schemes in countries where at least 80% of the population were insured. We used data collected through general practice and primary care clinics in high-income countries with universal insurance, because contact with the primary care systems tends to be at least as good as response rates for population-based surveys. No studies were excluded based on the level of height.

We used multiple routes for identifying and accessing data. We accessed publicly available population-based multi-country and national measurement surveys (e.g., Demographic and Health Surveys, and surveys identified via the Inter-University Consortium for Political and Social Research and European Health Interview & Health Examination Surveys Database) as well as the World Health Organization (WHO) STEPwise approach to Surveillance (STEPS) surveys. We requested identification and access to population-based data sources from ministries of health and other national health

agencies, via WHO and its regional offices. Requests were also sent via the World Heart Federation to its national partners. We made a similar request to the NCD Risk Factor Collaboration (NCD-RisC; www.ncdrisc.org), a worldwide network of health researchers and practitioners working on NCD risk factors.

To identify major sources not accessed through the above routes, we searched and reviewed published studies. Specifically, we searched Medline (via PubMed) for articles published between 1st January 1950 and 12th March 2013 using the search terms 'body size'[mh:noexp] OR 'body height'[mh:noexp] OR 'body weight'[mh:noexp] OR 'birth weight'[mh:noexp] OR 'overweight'[mh: noexp] OR 'obesity'[mh] OR 'thinness'[mh:noexp] OR 'Waist-Hip Ratio'[mh:noexp] or 'Waist Circum-ference'[mh:noexp] or 'body mass index' [mh:noexp]) AND ('Humans'[mh]) AND('1950'[PDAT]: '2013'[PDAT]) AND ('Health Surveys'[mh] OR 'Epidemiological Monitoring'[mh] OR 'Prevalence'[mh]) NOT Comment[ptyp] NOT Case Reports[ptyp].

Articles were screened according to the inclusion and exclusion criteria described above. The number of articles identified and retained is summarised in **Supplementary file 2**. As described above, we contacted the corresponding authors of all eligible studies and invited them to join NCD-RisC. We did similar searches for other cardio-metabolic risk factors including blood pressure, serum cholesterol, and blood glucose. All eligible studies were invited to join NCD-RisC and were requested to analyse data on all cardio-metabolic risk factors.

Anonymised individual record data from sources included in NCD-RisC were re-analysed by the Pooled Analysis and Writing Group or by data holders according to a common protocol. All re-analysed data sources included mean height in standard age groups (18 years, 19 years, 20–29 years, followed by 10 year age groups and 80+ years), as well as sample sizes and standard errors. All

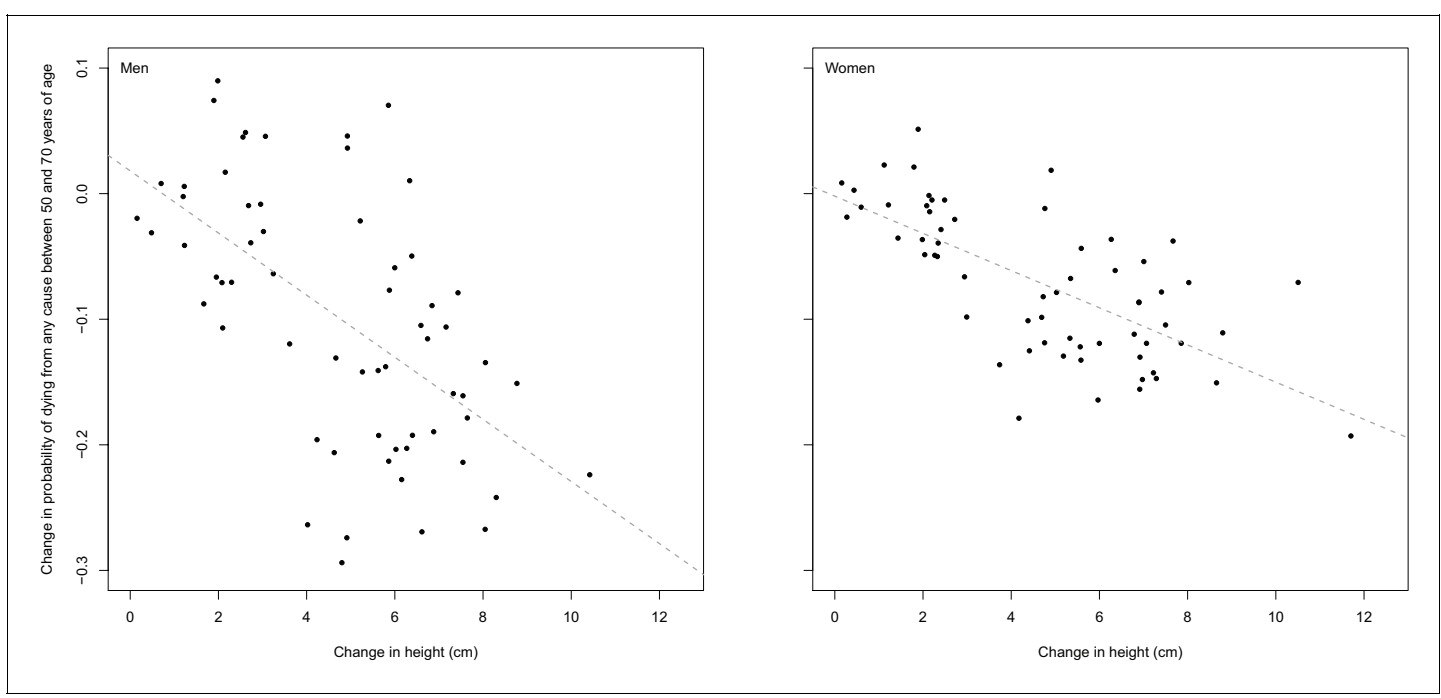

**Figure 11.** Association between change in probability of dying from any cause between 50 and 70 years of age and change in adult height by country for cohorts born between 1898 and 1946. Probability of death was calculated using a cohort life table. Mortality data were available for 1950 to 2013. The 1898 birth cohort is the first cohort whose mortality experience at 50–54 years of age was seen in the data, and the 1946 birth cohort the last cohort whose mortality experience at 65–69 years of age was seen in the data. The dotted line shows the linear association. The 62 countries included have vital registration that is >80% complete and have data on all-cause mortality for at least 30 cohorts. The countries are Argentina, Australia, Austria, Azerbaijan, Belarus, Belgium, Belize, Brazil, Bulgaria, Canada, Chile, China (Hong Kong SAR), Colombia, Costa Rica, Croatia, Cuba, Czech Republic, Denmark, Estonia, Finland, France, Germany, Greece, Guatemala, Hungary, Iceland, Ireland, Israel, Italy, Japan, Kazakhstan, Kyrgyzstan, Latvia, Lithuania, Luxembourg, Macedonia (TFYR), Malta, Mauritius, Mexico, Moldova, Netherlands, New Zealand, Norway, Poland, Portugal, Puerto Rico, Romania, Russian Federation, Slovakia, Slovenia, South Korea, Spain, Sweden, Switzerland, Trinidad and Tobago, Turkmenistan, Ukraine, United Kingdom, United States of America, Uruguay, Uzbekistan and Venezuela.

analyses incorporated appropriate sample weights and complex survey design when applicable. To ensure summaries were prepared according to the study protocol, the Pooled Analysis and Writing Group provided computer code to NCD-RisC members who requested assistance. We also recorded information about the study population, period of measurement and sampling approach. This information was used to establish that each data source was population-based, and to assess whether it covered the whole country, multiple subnational regions, or one or a small number of communities, and whether it was rural, urban, or combined. All submitted data were checked by at least two independent members of the Pooled Analysis and Writing Group to ensure that their sample selection met the inclusion criteria and that height was measured and not self-reported. Questions and clarifications about sample design and measurement method were discussed with the Collaborating Group members and resolved before data were incorporated in the database. We also extracted data from additional national health surveys, one subnational STEPS survey, and six MONICA sites from published reports.

We identified duplicate data sources by comparing studies from the same country and year. Additionally, NCD-RisC members received the list of all data sources in the database and were asked to ensure that the included data from their country met the inclusion criteria and that there were no duplicates. Data sources used in our analysis are listed in *Supplementary file 3*.

In this paper, we used data on height in adulthood (18 years of age and older) from the NCD-RisC database for participants born between 1896 and 1996. We used 1472 population-based data sources with measurements on over 18.6 million adults born between 1896 and 1996 whose height had been measured. We did not use data from the 1860–1895 cohorts because data on these early cohorts were available for only six countries (American Samoa, India, Japan, Norway, Switzerland and USA). We had data for 179 of the 200 countries for which estimates were made; these 179 countries covered 97% of the world's population. All countries had some data on people born after 1946 (second half of analysis period); 134 had data on people born between 1921 and 1945; and 72 had data on people born in 1920 or earlier. Across regions, there were between an average of 2.0 data sources per country in the Caribbean to 34 sources per country in high-income Asia Pacific. 1108 sources had data on men as well as women, 153 only on men, and 211 only on women.

## Statistical methods

The statistical method is described in detail elsewhere (*Danaei et al., 2011*; *Finucane et al., 2014*). In summary, the model had a hierarchical structure in which estimates of mean height for each country and year were nested in regional levels and trends, which were in turn nested in those of super-regions and worldwide. In this structure, estimates of mean height for each country and year were informed by its own data, if available, and by data from other years in the same country and in other countries, especially those in the same region with data for similar time periods. The hierarchical structure shares information to a greater degree when data are non-existent or weakly informative (e.g., because they have a small sample size), and to a lesser extent in data-rich countries and regions.

We used birth cohort as the time scale of analysis. We calculated the birth cohort for each observation by subtracting the mid-age of its age group from the year in which data were collected. We modelled trends in height by birth cohort as a combination of linear and non-linear trends, both with a hierarchical structure; the non-linear trend was specified using a second-order random walk (*Rue and Held, 2005*). The model also included a term that allowed each birth cohort's height to change as it aged, e.g., because there is gradual loss of height during ageing and because as a cohort ages those who survive may be taller. The model described by Finucane et al (*Finucane et al., 2014*) had used a cubic spline for age associations of risk factor levels. In practice, the estimated change in population mean height over age was linear with a small slope of over 0.2 cm shorter for men and 0.3 cm shorter for women with each decade of older age. Therefore, we used a linear specification for computational efficiency.

While all our data were from samples of the general population, 796 (54%) of data sources represented national populations, another 199 (14%) major sub-national regions (e.g., one or more provinces or regions of a country), and the remaining 477 (32%) one or a small number of communities. The model accounted for the fact that sub-national and community studies, while informative, might systematically differ from nationally representative ones, and also have larger variation relative to

the true values than national studies (e.g., see data from China, India, Japan and the UK in *Figure 6* and *Figure 7*).

We fitted the Bayesian model with the Markov chain Monte Carlo (MCMC) algorithm. We monitored mixing and convergence using trace plots and Brooks–Gelman–Rubin diagnostics (*Brooks and Gelman, 1998*). We obtained 5000 post burn-in samples from the posterior distribution of model parameters, used to obtain the posterior distribution of mean height. The reported credible intervals represent the 2.5th–97.5th percentiles of the posterior distribution. We report mean height at age 18 years for each birth cohort; heights at other ages are available from the authors. All analyses were done separately by sex because height and its trends over time may differ between men and women.

We tested how well our statistical model predicts missing values by removing data from 10% of countries with data (i.e., created the appearance of countries with no data where we actually had data). The countries whose data were withheld were randomly selected from the following three groups: data-rich (more than 25 cohorts of data, with at least five cohorts after 1960), data-poor (up to and including 12 cohorts of data for women and 8 cohorts for men), and average data availability (13 to 25 cohorts for women, 9 to 25 cohorts for men, or more than 25 cohorts in total with fewer than five after 1960). In total, there were 64 data-rich countries for women and 51 for men; 57 data-poor countries for women and 58 for men; and 56 countries for women and 60 for men that had average data availability. We fitted the model to the data from the remaining 90% of countries and made estimates of the held-out observations. We repeated the test five times, holding out a different subset of data in each repetition. We calculated the differences between the held-out data and the estimates. We also checked the 95% credible intervals of the estimates; in a model with good external predictive validity, 95% of held-out values would be included in the 95% credible intervals.

Our model performed extremely well; specifically, the estimates of mean height were unbiased as evidenced with median errors that were very close to zero globally, and less than ±0.2 cm in every subset of withheld data (*Supplementary file 4*). Even the 25th and 75th percentiles of errors rarely exceeded ±1 cm. Median absolute error was only about 0.5 cm, and did not exceed 1.0 cm in subsets of withheld data. The 95% credible intervals of estimated mean heights covered 97% of true data for both men and women, which implies good estimation of uncertainty; among subgroups of data, coverage was never < 90%.

## Acknowledgements

ME was awarded funding to carry out the research from the Wellcome Trust and Grand Challenges Canada. We thank Christina Banks, Quentin Hennocq, Dheeya Rizmie, and Yasaman Vali for assistance with data extraction. We thank WHO country and regional offices and World Heart Federation for support in data identification and access.

### NCD Risk Factor Collaboration (NCD-RisC)

Pooled Analysis and Writing (* equal contribution)
James Bentham (Imperial College London, UK)*; Mariachiara Di Cesare (Middlesex University, UK; Imperial College London, UK)*; Gretchen A Stevens (World Health Organization, Switzerland); Bin Zhou (Imperial College London, UK); Honor Bixby (Imperial College London, UK); Melanie Cowan (World Health Organization, Switzerland); Léa Fortunato (Imperial College London, UK); James E Bennett (Imperial College London, UK); Goodarz Danaei (Harvard T.H. Chan School of Public Health, USA); Kaveh Hajifathalian (Harvard T.H. Chan School of Public Health, USA); Yuan Lu (Harvard T.H. Chan School of Public Health, USA); Leanne M Riley (World Health Organization, Switzerland); Avula Laxmaiah (Indian Council of Medical Research, India); Vasilis Kontis (Imperial College London, UK); Christopher J Paciorek (University of California, Berkeley, USA); Elio Riboli (Imperial College London, UK); Majid Ezzati (Imperial College London, UK; WHO Collaborating Centre on NCD Surveillance and Epidemiology, UK).

Country and Regional Data (* equal contribution; listed alphabetically)
Ziad A Abdeen (Al-Quds University, Palestine)*; Zargar Abdul Hamid (Center for Diabetes and Endocrine Care, India)*; Niveen M Abu-Rmeileh (Birzeit University, Palestine)*; Benjamin Acosta-Cazares

(Instituto Mexicano del Seguro Social, Mexico)*; Robert Adams (The University of Adelaide, Australia)*; Wichai Aekplakorn (Mahidol University, Thailand)*; Carlos A Aguilar-Salinas (Instituto Nacional de Ciencias Médicas y Nutricion, Mexico)*; Charles Agyemang (University of Amsterdam, The Netherlands)*; Alireza Ahmadvand (Non-Communicable Diseases Research Center, Iran)*; Wolfgang Ahrens (Leibniz Institute for Prevention Research and Epidemiology - BIPS, Germany)*; Hazzaa M Al-Hazzaa (King Saud University, Saudi Arabia)*; Amani Rashed Al-Othman (Kuwait Institute for Scientific Research, Kuwait)*; Rajaa Al Raddadi (Ministry of Health, Saudi Arabia)*; Mohamed M Ali (World Health Organization Regional Office for the Eastern Mediterranean, Egypt)*; Ala'a Alkerwi (Luxembourg Institute of Health, Luxembourg)*; Mar Alvarez-Pedrerol (ISGlobal Centre for Research in Environmental Epidemiology, Spain)*; Eman Aly (World Health Organization Regional Office for the Eastern Mediterranean, Egypt)*; Philippe Amouyel (Lille University and Hospital, France)*; Antoinette Amuzu (London School of Hygiene & Tropical Medicine, UK)*; Lars Bo Andersen (Sogn and Fjordane University College, Norway)*; Sigmund A Anderssen (Norwegian School of Sport Sciences, Norway)*; Ranjit Mohan Anjana (Madras Diabetes Research Foundation, India)*; Hajer Aounallah-Skhiri (National Institute of Public Health, Tunisia)*; Inger Ariansen (Norwegian Institute of Public Health, Norway)*; Tahir Aris (Ministry of Health Malaysia, Malaysia)*; Nimmathota Arlappa (Indian Council of Medical Research, India)*; Dominique Arveiler (University of Strasbourg and Strasbourg University Hospital, France)*; Felix K Assah (University of Yaoundé 1, Cameroon)*; Mária Avdicová (Regional Authority of Public Health, Banska Bystrica, Slovakia)*; Fereidoun Azizi (Shahid Beheshti University of Medical Sciences, Iran)*; Bontha V Babu (Indian Council of Medical Research, India)*; Suhad Bahijri (King Abdulaziz University, Saudi Arabia)*; Nagalla Balakrishna (Indian Council of Medical Research, India)*; Piotr Bandosz (Medical University of Gdansk, Poland)*; José R Banegas (Universidad Autónoma de Madrid, Spain)*; Carlo M Barbagallo (University of Palermo, Italy)*; Alberto Barceló (Pan American Health Organization, USA)*; Amina Barkat (Mohammed V University de Rabat, Morocco)*; Mauro V Barros (University of Pernambuco, Brazil)*; Iqbal Bata (Dalhousie University, Canada)*; Anwar M Batieha (Jordan University of Science and Technology, Jordan)*; Rosangela L Batista (Federal University of Maranhao, Brazil)*; Louise A Baur (University of Sydney, Australia)*; Robert Beaglehole (University of Auckland, New Zealand)*; Habiba Ben Romdhane (University Tunis El Manar, Tunisia)*; Mikhail Benet (University Medical Science, Cuba)*; James E Bennett (Imperial College London, UK)*; Antonio Bernabe-Ortiz (Universidad Peruana Cayetano Heredia, Peru)*; Gailute Bernotiene (Lithuanian University of Health Sciences, Lithuania)*; Heloisa Bettiol (University of São Paulo, Brazil)*; Aroor Bhagyalaxmi (B. J. Medical College, India)*; Sumit Bharadwaj (Chirayu Medical College, India)*; Santosh K Bhargava (Sunder Lal Jain Hospital, India)*; Zaid Bhatti (The Aga Khan University, Pakistan)*; Zulfiqar A Bhutta (The Hospital for Sick Children, Canada; The Aga Khan University, Pakistan)*; Hongsheng Bi (Shandong University of Traditional Chinese Medicine, China)*; Yufang Bi (Shanghai Jiao-Tong University School of Medicine, China)*; Peter Bjerregaard (University of Southern Denmark, Denmark; University of Greenland, Greenland)*; Espen Bjertness (University of Oslo, Norway)*; Marius B Bjertness (University of Oslo, Norway)*; Cecilia Björkelund (University of Gothenburg, Sweden)*; Anneke Blokstra (National Institute for Public Health and the Environment, The Netherlands)*; Simona Bo (University of Turin, Italy)*; Martin Bobak (University College London, UK)*; Lynne M Boddy (Liverpool John Moores University, UK)*; Bernhard O Boehm (Nanyang Technological University, Singapore)*; Heiner Boeing (German Institute of Human Nutrition, Germany)*; Carlos P Boissonnet (CEMIC, Argentina)*; Vanina Bongard (Toulouse University School of Medicine, France)*; Pascal Bovet (Ministry of Health, Seychelles; University of Lausanne, Switzerland)*; Lutgart Braeckman (Ghent University, Belgium)*; Marjolijn CE Bragt (FrieslandCampina, Singapore)*; Imperia Brajkovich (Universidad Central de Venezuela, Venezuela)*; Francesco Branca (World Health Organization, Switzerland)*; Juergen Breckenkamp (Bielefeld University, Germany)*; Hermann Brenner (German Cancer Research Center, Germany)*; Lizzy M Brewster (University of Amsterdam, The Netherlands)*; Garry R Brian (The Fred Hollows Foundation New Zealand, New Zealand)*; Graziella Bruno (University of Turin, Italy)*; H.B(as) Bueno-de-Mesquita (National Institute for Public Health and the Environment, The Netherlands)*; Anna Bugge (University of Southern Denmark, Denmark)*; Con Burns (Cork Institute of Technology, Ireland)*; Antonio Cabrera de León (Universidad de La Laguna, Spain)*; Joseph Cacciottolo (University of Malta, Malta)*; Tilema Cama (Ministry of Health, Tonga)*; Christine Cameron (Canadian Fitness and Lifestyle Research Institute, Canada)*; José Camolas (Hospital Santa Maria, CHLN, Portugal)*; Günay Can (Istanbul University, Turkey)*; Ana Paula C Cândido (Universidade Federal de Juiz de Fora, Brazil)*; Vincenzo Capuano (Cardiologia di Mercato S.

Severino, Italy)*; Viviane C Cardoso (University of São Paulo, Brazil)*; Axel C Carlsson (Karolinska Institutet, Sweden)*; Maria J Carvalho (University of Porto, Portugal)*; Felipe F Casanueva (Santiago de Compostela University, Spain)*; Juan-Pablo Casas (University College London, UK)*; Carmelo A Caserta (Associazione Calabrese di Epatologia, Italy)*; Snehalatha Chamukuttan (India Diabetes Research Foundation, India)*; Angelique W Chan (Duke-NUS Graduate Medical School, Singapore)*; Queenie Chan (Imperial College London, UK)*; Himanshu K Chaturvedi (National Institute of Medical Statistics, India)*; Nishi Chaturvedi (University College London, UK)*; Chien-Jen Chen (Academia Sinica, Taiwan)*; Fangfang Chen (Capital Institute of Pediatrics, China)*; Huashuai Chen (Duke University, USA)*; Shuohua Chen (Kailuan General Hospital, China)*; Zhengming Chen (University of Oxford, UK)*; Ching-Yu Cheng (Duke-NUS Graduate Medical School, Singapore)*; Angela Chetrit (The Gertner Institute for Epidemiology and Health Policy Research, Israel)*; Arnaud Chiolero (Lausanne University Hospital, Switzerland)*; Shu-Ti Chiou (Ministry of Health and Welfare, Taiwan)*; Adela Chirita-Emandi (Victor Babes University of Medicine and Pharmacy Timisoara, Romania)*; Belong Cho (Seoul National University College of Medicine, South Korea)*; Yumi Cho (Korea Centers for Disease Control and Prevention, South Korea)*; Kaare Christensen (University of Southern Denmark, Denmark)*; Jerzy Chudek (Medical University of Silesia, Poland)*; Renata Cifkova (Charles University in Prague, Czech Republic)*; Frank Claessens (Katholieke Universiteit Leuven, Belgium)*; Els Clays (Ghent University, Belgium)*; Hans Concin (Agency for Preventive and Social Medicine, Austria)*; Cyrus Cooper (University of Southampton, UK)*; Rachel Cooper (University College London, UK)*; Tara C Coppinger (Cork Institute of Technology, Ireland)*; Simona Costanzo (IRCCS Istituto Neurologico Mediterraneo Neuromed, Italy)*; Dominique Cottel (Institut Pasteur de Lille, France)*; Chris Cowell (Westmead University of Sydney, Australia)*; Cora L Craig (Canadian Fitness and Lifestyle Research Institute, Canada)*; Ana B Crujeiras (CIBEROBN, Spain)*; Graziella D'Arrigo (National Council of Research, Italy)*; Eleonora d'Orsi (Federal University of Santa Catarina, Brazil)*; Jean Dallongeville (Institut Pasteur de Lille, France)*; Albertino Damasceno (Eduardo Mondlane University, Mozambique)*; Camilla T Damsgaard (University of Copenhagen, Denmark)*; Goodarz Danaei (Harvard TH Chan School of Public Health, USA)*; Rachel Dankner (The Gertner Institute for Epidemiology and Health Policy Research, Israel)*; Luc Dauchet (Lille University Hospital, France)*; Guy De Backer (Ghent University, Belgium)*; Dirk De Bacquer (Ghent University, Belgium)*; Giovanni de Gaetano (IRCCS Istituto Neurologico Mediterraneo Neuromed, Italy)*; Stefaan De Henauw (Ghent University, Belgium)*; Delphine De Smedt (Ghent University, Belgium)*; Mohan Deepa (Madras Diabetes Research Foundation, India)*; Alexander D Deev (National Research Centre for Preventive Medicine, Russia)*; Abbas Dehghan (Erasmus Medical Center Rotterdam, The Netherlands)*; Hélène Delisle (University of Montreal, Canada)*; Francis Delpeuch (Institut de Recherche pour le Développement, France)*; Valérie Deschamps (French Public Health Agency, France)*; Klodian Dhana (Erasmus Medical Center Rotterdam, The Netherlands)*; Augusto F Di Castelnuovo (IRCCS Istituto Neurologico Mediterraneo Neuromed, Italy)*; Juvenal Soares Dias-da-Costa (Universidade do Vale do Rio dos Sinos, Brazil)*; Alejandro Diaz (National Council of Scientific and Technical Research, Argentina)*; Shirin Djalalinia (Non-Communicable Diseases Research Center, Iran)*; Ha TP Do (National Institute of Nutrition, Vietnam)*; Annette J Dobson (University of Queensland, Australia)*; Chiara Donfrancesco (Istituto Superiore di Sanità, Italy)*; Silvana P Donoso (Universidad de Cuenca, Ecuador)*; Angela Döring (Helmholtz Zentrum München, Germany)*; Kouamelan Doua (Ministère de la Santé et de la Lutte Contre le Sida, Côte d'Ivoire)*; Wojciech Drygas (The Cardinal Wyszynski Institute of Cardiology, Poland)*; Vilnis Dzerve (University of Latvia, Latvia)*; Eruke E Egbagbe (University of Benin, Nigeria)*; Robert Eggertsen (University of Gothenburg, Sweden)*; Ulf Ekelund (Norwegian School of Sport Sciences, Norway)*; Jalila El Ati (National Institute of Nutrition and Food Technology, Tunisia)*; Paul Elliott (Imperial College London, UK)*; Reina Engle-Stone (University of California Davis, USA)*; Rajiv T Erasmus (University of Stellenbosch, South Africa)*; Cihangir Erem (Karadeniz Technical University, Turkey)*; Louise Eriksen (University of Southern Denmark, Denmark)*; Jorge Escobedo-de la Peña (Instituto Mexicano del Seguro Social, Mexico)*; Alun Evans (The Queen's University of Belfast, UK)*; David Faeh (University of Zurich, Switzerland)*; Caroline H Fall (University of Southampton, UK)*; Farshad Farzadfar (Tehran University of Medical Sciences, Iran)*; Francisco J Felix-Redondo (Centro de Salud Villanueva Norte, Spain)*; Trevor S Ferguson (The University of the West Indies, Jamaica)*; Daniel Fernández-Bergés (Hospital Don Benito-Villanueva de la Serena, Spain)*; Daniel Ferrante (Ministry of Health, Argentina)*; Marika Ferrari (Council for Agricultural Research and Economics, Italy)*; Catterina Ferreccio (Pontificia Universidad Católica de Chile, Chile)

*; Jean Ferrieres (Toulouse University School of Medicine, France)*; Joseph D Finn (University of Manchester, UK)*; Krista Fischer (University of Tartu, Estonia)*; Eric Monterubio Flores (Instituto Nacional de Salud Pública, Mexico)*; Bernhard Föger (Agency for Preventive and Social Medicine, Austria)*; Leng Huat Foo (Universiti Sains Malaysia, Malaysia)*; Ann-Sofie Forslund (Luleå University, Sweden)*; Maria Forsner (Dalarna University, Sweden)*; Stephen P Fortmann (Stanford University, USA)*; Heba M Fouad (World Health Organization Regional Office for the Eastern Mediterranean, Egypt)*; Damian K Francis (The University of the West Indies, Jamaica)*; Maria do Carmo Franco (Federal University of São Paulo, Brazil)*; Oscar H Franco (Erasmus Medical Center Rotterdam, The Netherlands)*; Guillermo Frontera (Hospital Universitario Son Espases, Spain)*; Flavio D Fuchs (Hospital de Clinicas de Porto Alegre, Brazil)*; Sandra C Fuchs (Universidade Federal do Rio Grande do Sul, Brazil)*; Yuki Fujita (Kindai University Faculty of Medicine, Japan)*; Takuro Furusawa (Kyoto University, Japan)*; Zbigniew Gaciong (Medical University of Warsaw, Poland)*; Mihai Gafencu (Victor Babes University of Medicine and Pharmacy Timisoara, Romania)*; Dickman Gareta (University of KwaZulu-Natal, South Africa)*; Sarah P Garnett (University of Sydney, Australia)*; Jean-Michel Gaspoz (Geneva University Hospitals, Switzerland)*; Magda Gasull (CIBER en Epidemiología y Salud Pública, Spain)*; Louise Gates (Australian Bureau of Statistics, Australia)*; Johanna M Geleijnse (Wageningen University, The Netherlands)*; Anoosheh Ghasemian (Non-Communicable Diseases Research Center, Iran)*; Simona Giampaoli (Istituto Superiore di Sanità, Italy)*; Francesco Gianfagna (University of Insubria, Italy)*; Jonathan Giovannelli (Lille University Hospital, France)*; Aleksander Giwercman (Lund University, Sweden)*; Rebecca A Goldsmith (Nutrition Department, Ministry of Health, Israel)*; Helen Gonçalves (Federal University of Pelotas, Brazil)*; Marcela Gonzalez Gross (Universidad Politécnica de Madrid, Spain)*; Juan P González Rivas (The Andes Clinic of Cardio-Metabolic Studies, Venezuela)*; Mariano Bonet Gorbea (National Institute of Hygiene, Epidemiology and Microbiology, Cuba)*; Frederic Gottrand (Université de Lille 2, France)*; Sidsel Graff-Iversen (Norwegian Institute of Public Health, Norway)*; Dušan Grafnetter (Institute for Clinical and Experimental Medicine, Czech Republic)*; Aneta Grajda (Children's Memorial Health Institute, Poland)*; Maria G Grammatikopoulou (Alexander Technological Educational Institute, Greece)*; Ronald D Gregor (Dalhousie University, Canada)*; Tomasz Grodzicki (Jagiellonian University Medical College, Poland)*; Anders Grøntved (University of Southern Denmark, Denmark)*; Grabriella Gruden (University of Turin, Italy)*; Vera Grujic (University of Novi Sad, Serbia)*; Dongfeng Gu (National Center of Cardiovascular Diseases, China)*; Emanuela Gualdi-Russo (University of Ferrara, Italy)*; Ong Peng Guan (Singapore Eye Research Institute, Singapore)*; Vilmundur Gudnason (Icelandic Heart Association, Iceland)*; Ramiro Guerrero (Universidad Icesi, Colombia)*; Idris Guessous (Geneva University Hospitals, Switzerland)*; Andre L Guimaraes (State University of Montes Claros, Brazil)*; Martin C Gulliford (King's College London, UK)*; Johanna Gunnlaugsdottir (Icelandic Heart Association, Iceland)*; Marc Gunter (Imperial College London, UK)*; Xiuhua Guo (Capital Medical University, China)*; Yin Guo (Capital Medical University, China)*; Prakash C Gupta (Healis - Sekhsaria Institute for Public Health, India)*; Oye Gureje (University of Ibadan, Nigeria)*; Beata Gurzkowska (Children's Memorial Health Institute, Poland)*; Laura Gutierrez (Institute for Clinical Effectiveness and Health Policy, Argentina)*; Felix Gutzwiller (University of Zurich, Switzerland)*; Jytte Halkjær (Danish Cancer Society Research Centre, Denmark)*; Ian R Hambleton (The University of the West Indies, Barbados)*; Rebecca Hardy (University College London, UK)*; Rachakulla Hari Kumar (Indian Council of Medical Research, India)*; Jun Hata (Kyushu University, Japan)*; Alison J Hayes (University of Sydney, Australia)*; Jiang He (Tulane University, USA)*; Marleen Elisabeth Hendriks (Academic Medical Center of University of Amsterdam, The Netherlands)*; Leticia Hernandez Cadena (National Institute of Public Health, Mexico)*; Sauli Herrala (Oulu University Hospital, Finland)*; Ramin Heshmat (Chronic Diseases Research Center, Iran)*; Ilpo Tapani Hihtaniemi (Imperial College London, UK)*; Sai Yin Ho (University of Hong Kong, China), Suzanne C Ho (The Chinese University of Hong Kong, China)*; Michael Hobbs (University of Western Australia, Australia)*; Albert Hofman (Erasmus Medical Center Rotterdam, The Netherlands)*; Claudia M Hormiga (Fundación Oftalmológica de Santander, Colombia)*; Bernardo L Horta (Universidade Federal de Pelotas, Brazil)*; Leila Houti (University of Oran 1, Algeria)*; Christina Howitt (The University of the West Indies, Barbados)*; Thein Thein Htay (Independent Public Health Specialist, Myanmar)*; Aung Soe Htet (University of Oslo, Norway)*; Maung Maung Than Htike (International Realtions Division, Nay Pyi Taw)*; Yonghua Hu (Peking University Health Science Center, China)*; Abdullatif Husseini (Birzeit University, Palestine)*; Chinh Nguyen Huu (National Institute of Nutrition, Vietnam)*; Inge Huybrechts (International Agency for Research on

Cancer, France)*; Nahla Hwalla (American University of Beirut, Lebanon)*; Licia Iacoviello (IRCCS Istituto Neurologico Mediterraneo Neuromed, Italy)*; Anna G Iannone (Cardiologia di Mercato S. Severino, Italy)*; Mohsen M Ibrahim (Cairo University, Egypt)*; Nayu Ikeda (National Institute of Health and Nutrition, Japan)*; M Arfan Ikram (Erasmus Medical Center Rotterdam, The Netherlands)*; Vilma E Irazola (Institute for Clinical Effectiveness and Health Policy, Argentina)*; Muhammad Islam (Aga Khan University, Pakistan)*; Vanja Ivkovic (UHC Zagreb, Croatia)*; Masanori Iwasaki (Niigata University, Japan)*; Rod T Jackson (University of Auckland, New Zealand)*; Jeremy M Jacobs (Hadassah University Medical Center, Israel)*; Tazeen Jafar (Duke-NUS Graduate Medical School, Singapore)*; Kazi M Jamil (Kuwait Institute for Scientific Research, Kuwait)*; Konrad Jamrozik (University of Adelaide, Australia; deceased)*; Imre Janszky (Norwegian University of Science and Technology, Norway)*; Grazyna Jasienska (Jagiellonian University Medical College, Poland)*; Bojan Jelakovic (University of Zagreb School of Medicine, Croatia)*; Chao Qiang Jiang (Guangzhou 12th Hospital, China)*; Michel Joffres (Simon Fraser University, Canada)*; Mattias Johansson (International Agency for Research on Cancer, France)*; Jost B Jonas (Ruprecht-Karls-University of Heidelberg, Germany)*; Torben Jørgensen (Research Centre for Prevention and Health, Denmark)*; Pradeep Joshi (World Health Organization Country Office, India)*; Anne Juolevi (National Institute for Health and Welfare, Finland)*; Gregor Jurak (University of Ljubljana, Slovenia)*; Vesna Jureša (University of Zagreb, Croatia)*; Rudolf Kaaks (German Cancer Research Center, Germany)*; Anthony Kafatos (University of Crete, Greece)*; Ofra Kalter-Leibovici (The Gertner Institute for Epidemiology and Health Policy Research, Israel)*; Efthymios Kapantais (Hellenic Medical Association for Obesity, Greece)*; Amir Kasaeian (Tehran University of Medical Science, Iran)*; Joanne Katz (Johns Hopkins Bloomberg School of Public Health, USA)*; Prabhdeep Kaur (National Institute of Epidemiology, India)*; Maryam Kavousi (Erasmus Medical Center Rotterdam, The Netherlands)*; Ulrich Keil (University of Münster, Germany)*; Lital Keinan Boker ( Israel Center for Disease Control, Israel)*; Sirkka Keinänen-Kiukaanniemi (Oulu University Hospital, Finland)*; Roya Kelishadi (Research Institute for Primordial Prevention of Non Communicable Disease, Iran)*; Han CG Kemper (VU University Medical Center, The Netherlands)*; Andre P Kengne (South African Medical Research Council, South Africa)*; Mathilde Kersting (Research Institute of Child Nutrition, Germany)*; Timothy Key (University of Oxford, UK)*; Yousef Saleh Khader (Jordan University of Science and Technology, Jordan)*; Davood Khalili (Shahid Beheshti University of Medical Sciences, Iran)*; Young-Ho Khang (Seoul National University, South Korea)*; Kay-Tee H Khaw (University of Cambridge, UK)*; Ilse MSL Khouw (FrieslandCampina, Singapore)*; Stefan Kiechl (Medical University Innsbruck, Austria)*; Japhet Killewo (Muhimbili University of Health and Allied Sciences, Tanzania)*; Jeongseon Kim (National Cancer Center, South Korea), Jeannette Klimont (Statistics Austria, Austria)*; Jurate Klumbiene (Lithuanian University of Health Sciences, Lithuania)*; Bhawesh Koirala (B P Koirala Institute of Health Sciences, Nepal)*; Elin Kolle (Norwegian School of Sport Sciences, Norway)*; Patrick Kolsteren (Institute of Tropical Medicine, Belgium)*; Paul Korrovits (Tartu University Clinics, Estonia)*; Seppo Koskinen (National Institute for Health and Welfare, Finland)*; Katsuyasu Kouda (Kindai University Faculty of Medicine, Japan)*; Slawomir Koziel (Polish Academy of Sciences Anthropology Unit in Wroclaw, Poland)*; Wolfgang Kratzer (University Hospital Ulm, Germany)*; Steinar Krokstad (Norwegian University of Science and Technology, Norway)*; Daan Kromhout (Wageningen University, The Netherlands)*; Herculina S Kruger (North-West University, South Africa)*; Ruzena Kubinova (National Institute of Public Health, Czech Republic)*; Urho M Kujala (University of Jyväskylä, Finland)*; Krzysztof Kula (Medical University of Lodz, Poland)*; Zbigniew Kulaga (The Children's Memorial Health Institute, Poland)*; R Krishna Kumar (Amrita Institute of Medical Sciences, India)*; Pawel Kurjata (The Cardinal Wyszynski Institute of Cardiology, Poland)*; Yadlapalli S Kusuma (All India Institute of Medical Sciences, India)*; Kari Kuulasmaa (National Institute for Health and Welfare, Finland)*; Catherine Kyobutungi (African Population and Health Research Center, Kenya)*; Fatima Zahra Laamiri (Higher Institute of Nursing Professions and Technical Health, Morocco)*; Tiina Laatikainen (National Institute for Health and Welfare, Finland)*; Carl Lachat (Ghent University, Belgium)*; Youcef Laid (National Institute of Public Health of Algeria, Algeria)*; Tai Hing Lam (University of Hong Kong, China)*; Orlando Landrove (Ministerio de Salud Pública, Cuba)*; Vera Lanska (Institute for Clinical and Experimental Medicine, Czech Republic)*; Georg Lappas (Sahlgrenska Academy, Sweden)*; Bagher Larijani (Endocrinology and Metabolism Research Center, Iran)*; Lars E Laugsand (Norwegian University of Science and Technology, Norway)*; Avula Laxmaiah (Indian Council of Medical Research, India)*; Khanh Le Nguyen Bao (National Institute of Nutrition, Vietnam)*; Tuyen D Le (National Institute of Nutrition, Vietnam)*;

Catherine Leclercq (Food and Agriculture Organization, Italy)*; Jeannette Lee (National University of Singapore, Singapore)*; Jeonghee Lee (National Cancer Center, South Korea)*; Terho Lehtimäki (Tampere University Hospital, Finland)*; Rampal Lekhraj (Universiti Putra Malaysia, Malaysia)*; Luz M León-Muñoz (Universidad Autónoma de Madrid, Spain)*; Yanping Li (Harvard TH Chan School of Public Health, USA)*; Christa L Lilly (West Virginia University, USA)*; Wei-Yen Lim (National University of Singapore, Singapore)*; M Fernanda Lima-Costa (Oswaldo Cruz Foundation Rene Rachou Research Institute, Brazil)*; Hsien-Ho Lin (National Taiwan University, Taiwan)*; Xu Lin (University of Chinese Academy of Sciences, China)*; Allan Linneberg (Research Centre for Prevention and Health, Denmark)*; Lauren Lissner (University of Gothenburg, Sweden)*; Mieczyslaw Litwin (The Children's Memorial Health Institute, Poland)*; Jing Liu (Beijing Anzhen Hospital, Capital Medical University, China)*; Roberto Lorbeer (University Medicine Greifswald, Germany)*; Paulo A Lotufo (University of São Paulo, Brazil)*; José Eugenio Lozano (Consejería de Sanidad Junta de Castilla y León, Spain)*; Dalia Luksiene (Lithuanian University of Health Sciences, Lithuania)*; Annamari Lundqvist (National Institute for Health and Welfare, Finland)*; Nuno Lunet (Universidade do Porto, Portugal)*; Per Lytsy (University of Uppsala, Sweden)*; Guansheng Ma (Peking University, China)*; Jun Ma (Peking University, China)*; George LL Machado-Coelho (Universidade Federal de Ouro Preto, Brazil)*; Suka Machi (The Jikei University School of Medicine, Japan)*; Stefania Maggi (National Research Council, Italy)*; Dianna J Magliano (Baker IDI Heart and Diabetes Institute, Australia)*; Bernard Maire (Institut de Recherche pour le Développement, France)*; Marcia Makdisse (Hospital Israelita Albert Einstein, Brazil)*; Reza Malekzadeh (Tehran University of Medical Sciences, Iran)*; Rahul Malhotra (Duke-NUS Graduate Medical School, Singapore)*; Kodavanti Mallikharjuna Rao (Indian Council of Medical Research, India)*; Sofia Malyutina (Institute of Internal and Preventive Medicine, Russia)*; Yannis Manios (Harokopio University, Greece)*; Jim I Mann (University of Otago, New Zealand)*; Enzo Manzato (University of Padova, Italy)*; Paula Margozzini (Pontificia Universidad Católica de Chile, Chile)*; Oonagh Markey (University of Reading, UK)*; Pedro Marques-Vidal (Lausanne University Hospital, Switzerland)*; Jaume Marrugat (Institut Hospital del Mar d'Investigacions Mèdiques, Spain)*; Yves Martin-Prevel (Institut de Recherche pour le Développement, France)*; Reynaldo Martorell (Emory University, USA)*; Shariq R Masoodi (Sher-i-Kashmir Institute of Medical Sciences, India)*; Ellisiv B Mathiesen (UiT The Arctic University of Norway, Norway)*; Tandi E Matsha (Cape Peninsula University of Technology, South Africa)*; Artur Mazur (University of Rzeszow, Poland)*; Jean Claude N Mbanya (University of Yaoundé 1, Cameroon)*; Shelly R McFarlane (The University of the West Indies, Jamaica)*; Stephen T McGarvey (Brown University, USA)*; Martin McKee (London School of Hygiene & Tropical Medicine, UK)*; Stela McLachlan (University of Edinburgh, UK)*; Rachael M McLean (University of Otago, New Zealand)*; Breige A McNulty (University College Dublin, Ireland)*; Safiah Md Yusof (Universiti Teknologi MARA, Malaysia)*; Sounnia Mediene-Benchekor (University of Oran 1, Algeria)*; Aline Meirhaeghe (Institut National de la Santé et de la Recherche Médicale, France)*; Christa Meisinger (Helmholtz Zentrum München, Germany)*; Ana Maria B Menezes (Universidade Federal de Pelotas, Brazil)*; Gert BM Mensink (Robert Koch Institute, Germany)*; Indrapal I Meshram (Indian Council of Medical Research, India)*; Andres Metspalu (University of Tartu, Estonia)*; Jie Mi (Capital Institute of Pediatrics, China)*; Kim F Michaelsen (University of Copenhagen, Denmark)*; Kairit Mikkel (University of Tartu, Estonia)*; Jody C Miller (University of Otago, New Zealand)*; Juan Francisco Miquel (Pontificia Universidad Católica de Chile, Chile)*; J Jaime Miranda (Universidad Peruana Cayetano Heredia, Peru)*; Marjeta Mišigoj-Durakovic (University of Zagreb, Croatia)*; Mostafa K Mohamed (Ain Shams University, Egypt)*; Kazem Mohammad (Tehran University of Medical Sciences, Iran)*; Noushin Mohammadifard (Isfahan Cardiovascular Research Center, Iran)*; Viswanathan Mohan (Madras Diabetes Research Foundation, India)*; Muhammad Fadhli Mohd Yusoff (Ministry of Health Malaysia, Malaysia)*; Drude Molbo (University of Copenhagen, Denmark)*; Niels C Møller (University of Southern Denmark, Denmark)*; Dénes Molnár (University of Pécs, Hungary)*; Charles K Mondo (Mulago Hospital, Uganda)*; Eric A Monterrubio (Instituto Nacional de Salud Pública, Mexico)*; Kotsedi Daniel K Monyeki (University of Limpopo, South Africa)*; Leila B Moreira (Universidade Federal do Rio Grande do Sul, Brazil)*; Alain Morejon (University Medical Science, Cuba)*; Luis A Moreno (Universidad de Zaragoza, Spain)*; Karen Morgan (RCSI Dublin, Ireland)*; Erik Lykke Mortensen (University of Copenhagen, Denmark)*; George Moschonis (Harokopio University, Greece)*; Malgorzata Mossakowska (International Institute of Molecular and Cell Biology, Poland)*; Aya Mostafa (Ain Shams University, Egypt)*; Jorge Mota (University of Porto, Portugal)*; Mohammad Esmaeel Motlagh (Ahvaz Jundishapur University of Medical Sciences, Iran)*; Jorge Motta (Gorgas

Memorial Institute of Public Health, Panama)*; Thet Thet Mu (Department of Public Health, Myanmar)*; Maria Lorenza Muiesan (University of Brescia, Italy)*; Martina Müller-Nurasyid (Helmholtz Zentrum München, Germany)*; Neil Murphy (Imperial College London, UK)*; Jaakko Mursu (University of Eastern Finland, Finland)*; Elaine M Murtagh (Mary Immaculate College, Ireland)*; Kamarul Imran Musa (Universiti Sains Malaysia, Kota Bharu, Malaysia)*; Vera Musil (University of Zagreb, Croatia)*; Gabriele Nagel (Ulm University, Germany)*; Harunobu Nakamura (Kobe University, Japan)*; Jana Námešná (Regional Authority of Public Health, Banska Bystrica, Slovakia)*; Ei Ei K Nang (National University of Singapore, Singapore)*; Vinay B Nangia (Suraj Eye Institute, India)*; Martin Nankap (Helen Keller International, Cameroon)*; Sameer Narake (Healis - Sekhsaria Institute for Public Health, India)*; Eva Maria Navarrete-Muñoz (CIBER en Epidemiología y Salud Pública, Spain)*; William A Neal (West Virginia University, USA)*; Ilona Nenko (Jagiellonian University Medical College, Poland)*; Martin Neovius (Karolinska Institutet, Sweden)*; Flavio Nervi (Pontificia Universidad Católica de Chile, Chile)*; Hannelore K Neuhauser (Robert Koch Institute, Germany)*; Nguyen D Nguyen (University of Pharmacy and Medicine of Ho Chi Minh City, Vietnam)*; Quang Ngoc Nguyen (Hanoi Medical University, Vietnam)*; Ramfis E Nieto-Martínez (Universidad Centro-Occidental Lisandro Alvarado, Venezuela)*; Guang Ning (Shanghai Jiao-Tong University School of Medicine, China)*; Toshiharu Ninomiya (Kyushu University, Japan)*; Sania Nishtar (Heartfile, Pakistan)*; Marianna Noale (National Research Council, Italy)*; Teresa Norat (Imperial College London, UK)*; Davide Noto (University of Palermo, Italy)*; Mohannad Al Nsour (Eastern Mediterranean Public Health Network, Jordan)*; Dermot O'Reilly (The Queen's University of Belfast, UK)*; Kyungwon Oh (Korea Centers for Disease Control and Prevention, South Korea)*; Iman H Olayan (Kuwait Institute for Scientific Research, Kuwait)*; Maria Teresa Anselmo Olinto (University of Vale do Rio dos Sinos, Brazil)*; Maciej Oltarzewski (National Food and Nutrition Institute, Poland), Mohd A Omar (Ministry of Health Malaysia, Malaysia)*; Altan Onat (Istanbul University, Turkey)*; Pedro Ordunez (Pan American Health Organization, USA)*; Ana P Ortiz (University of Puerto Rico, Puerto Rico)*; Merete Osler (Research Center for Prevention and Health, Denmark)*; Clive Osmond (MRC Lifecourse Epidemiology Unit, UK)*; Sergej M Ostojic (University of Novi Sad, Serbia)*; Johanna A Otero (Fundación Oftalmológica de Santander, Colombia)*; Kim Overvad (Aarhus University, Denmark)*; Ellis Owusu-Dabo (Kwame Nkrumah University of Science and Technology, Ghana)*; Fred Michel Paccaud (Institute for Social and Preventive Medicine, Switzerland)*; Cristina Padez (University of Coimbra, Portugal)*; Elena Pahomova (University of Latvia, Latvia)*; Andrzej Pajak (Jagiellonian University Medical College, Poland)*; Domenico Palli (Cancer Prevention and Research Institute, Italy)*; Alberto Palloni (University of Wisconsin-Madison, USA)*; Luigi Palmieri (Istituto Superiore di Sanità, Italy)*; Songhomitra Panda-Jonas (Ruprecht-Karls-University of Heidelberg, Germany)*; Francesco Panza (University of Bari, Italy)*; Winsome R Parnell (University of Otago, New Zealand)*; Mahboubeh Parsaeian (Tehran University of Medical Sciences, Iran)*; Ivan Pecin (University of Zagreb, Croatia)*; Mangesh S Pednekar (Healis - Sekhsaria Institute for Public Health, India)*; Petra H Peeters (University Medical Center Utrecht, The Netherlands)*; Sergio Viana Peixoto (Oswaldo Cruz Foundation Rene Rachou Research Institute, Brazil)*; Markku Peltonen (National Institute for Health and Welfare, Finland)*; Alexandre C Pereira (Heart Institute, Brazil)*; Cynthia M Pérez (University of Puerto Rico, Puerto Rico)*; Annette Peters (Helmholtz Zentrum München, Germany)*; Janina Petkeviciene (Lithuanian University of Health Sciences, Lithuania)*; Niloofar Peykari (Non-Communicable Diseases Research Center, Iran)*; Son Thai Pham (Vietnam National Heart Institute, Vietnam)*; Iris Pigeot (Leibniz Institute for Prevention Research and Epidemiology - BIPS, Germany)*; Hynek Pikhart (University College London, UK)*; Aida Pilav (Federal Ministry of Health, Bosnia and Herzegovina)*; Lorenza Pilotto (Cardiovascular Prevention Centre, Italy)*; Francesco Pistelli (University Hospital of Pisa, Italy)*; Freda Pitakaka (University of New South Wales, Australia)*; Aleksandra Piwonska (The Cardinal Wyszynski Institute of Cardiology, Poland)*; Pedro Plans-Rubió (Public Health Agency of Catalonia, Spain)*; Bee Koon Poh (Universiti Kebangsaan Malaysia, Malaysia)*; Miquel Porta (Institut Hospital del Mar d'Investigacions Mèdiques, Spain)*; Marileen LP Portegies (Erasmus Medical Center Rotterdam, The Netherlands)*; Dimitrios Poulimeneas (Alexander Technological Educational Institute, Greece)*; Rajendra Pradeepa (Madras Diabetes Research Foundation, India)*; Mathur Prashant (Indian Council of Medical Research, India)*; Jacqueline F Price (University of Edinburgh, UK)*; Maria Puiu (Victor Babes University of Medicine and Pharmacy Timisoara, Romania)*; Margus Punab (Tartu University Clinics, Estonia), Radwan F Qasrawi (Al-Quds University, Palestine)*; Mostafa Qorbani (Alborz University of Medical Sciences, Iran)*; Tran Quoc Bao (Ministry of Health, Vietnam)*; Ivana Radic (University of

Novi Sad, Serbia)*; Ricardas Radisauskas (Lithuanian University of Health Sciences, Lithuania)*; Mahmudur Rahman (Institute of Epidemiology Disease Control and Research, Bangladesh)*; Olli Raitakari (Turku University Hospital, Finland)*; Manu Raj (Amrita Institute of Medical Sciences, India)*; Sudha Ramachandra Rao (National Institute of Epidemiology, India)*; Ambady Ramachandran (India Diabetes Research Foundation, India)*; Jacqueline Ramke (University of New South Wales, Australia)*; Rafel Ramos (Institut Universitari d'Investigació en Atenció Primària Jordi Gol, Spain)*; Sanjay Rampal (University of Malaya, Malaysia)*; Finn Rasmussen (Karolinska Institutet, Sweden)*; Josep Redon (University of Valencia, Spain)*; Paul Ferdinand M Reganit (University of the Philippines, Philippines)*; Robespierre Ribeiro (Minas Gerais State Secretariat for Health, Brazil)*; Elio Riboli (Imperial College London, UK)*; Fernando Rigo (Health Center San Agustín, Spain)*; Tobias F Rinke de Wit (PharmAccess Foundation, The Netherlands)*; Raphael M Ritti-Dias (Hospital Israelita Albert Einstein, Brazil)*; Juan A Rivera (Instituto Nacional de Salud Pública, Mexico)*; Sian M Robinson (University of Southampton, UK)*; Cynthia Robitaille (Public Health Agency of Canada, Canada)*; Fernando Rodríguez-Artalejo (Universidad Autónoma de Madrid, Spain)*; María del Cristo Rodriguez-Perez (Canarian Health Service, Spain)*; Laura A Rodríguez-Villamizar (Universidad Industrial de Santander, Colombia)*; Rosalba Rojas-Martinez (Instituto Nacional de Salud Pública, Mexico)*; Nipa Rojroongwasinkul (Mahidol University, Thailand)*; Dora Romaguera (CIBEROBN, Spain)*; Kimmo Ronkainen (University of Eastern Finland, Finland)*; Annika Rosengren (University of Gothenburg, Sweden)*; Ian Rouse (Fiji National University, Fiji)*; Adolfo Rubinstein (Institute for Clinical Effectiveness and Health Policy, Argentina)*; Frank J Rühli (University of Zurich, Switzerland)*; Ornelas Rui (University of Madeira, Portugal)*; Blanca Sandra Ruiz-Betancourt (Instituto Mexicano del Seguro Social, Mexico)*; Andrea RV Russo Horimoto (Heart Institute, Brazil)*; Marcin Rutkowski (Medical University of Gdansk, Poland)*; Charumathi Sabanayagam (Singapore Eye Research Institute, Singapore)*; Harshpal S Sachdev (Sitaram Bhartia Institute of Science and Research, India)*; Olfa Saidi (Faculty of medicine of Tunis, Tunisia)*; Benoit Salanave (French Public Health Agency, France)*; Eduardo Salazar Martinez (National Institute of Public Health, Mexico)*; Veikko Salomaa (National Institute for Health and Welfare, Finland)*; Jukka T Salonen (University of Helsinki, Finland)*; Massimo Salvetti (University of Brescia, Italy)*; Jose Sánchez-Abanto (National Institute of Health, Peru)*; Sandjaja (Ministry of Health, Indonesia); Susana Sans (Catalan Department of Health, Spain)*; Diana A Santos (Universidade de Lisboa, Portugal)*; Osvaldo Santos (Institute of Preventive Medicine and Public Health, Portugal)*; Renata Nunes dos Santos (University of Sao Paulo Clinics Hospital, Brazil)*; Rute Santos (University of Porto, Portugal)*; Jouko L Saramies (South Karelia Social and Health Care District, Finland)*; Luis B Sardinha (Universidade de Lisboa, Portugal)*; Nizal Sarrafzadegan (Isfahan Cardiovascular Research Center, Iran)*; Kai-Uwe Saum (German Cancer Research Center, Germany)*; Savvas C Savva (Research and Education Institute of Child Health, Cyprus)*; Marcia Scazufca (University of Sao Paulo Clinics Hospital, Brazil)*; Angelika Schaffrath Rosario (Robert Koch Institute, Germany)*; Herman Schargrodsky (Hospital Italiano de Buenos Aires, Argentina)*; Anja Schienkiewitz (Robert Koch Institute, Germany)*; Ida Maria Schmidt (Rigshospitalet, Denmark)*; Ione J Schneider (Federal University of Santa Catarina, Brazil)*; Constance Schultsz (Academic Medical Center of University of Amsterdam, The Netherlands)*; Aletta E Schutte (MRC North-West University, South Africa)*; Aye Aye Sein (Ministry of Health, Myanmar)*; Abhijit Sen (Norwegian University of Science and Technology, Norway)*; Idowu O Senbanjo (Lagos State University College of Medicine, Nigeria)*; Sadaf G Sepanlou (Digestive Diseases Research Institute, Iran)*; Svetlana A Shalnova (National Research Centre for Preventive Medicine, Russia)*; Sanjib K Sharma (B P Koirala Institute of Health Sciences, Nepal)*; Jonathan E Shaw (Baker IDI Heart and Diabetes Institute, Australia)*; Kenji Shibuya (The University of Tokyo, Japan)*; Dong Wook Shin (Seoul National University College of Medicine, South Korea)*; Youchan Shin (Singapore Eye Research Institute, Singapore)*; Rahman Shiri (Finnish Institute of Occupational Health, Finland)*; Rosalynn Siantar (Singapore Eye Research Institute, Singapore)*; Abla M Sibai (American University of Beirut, Lebanon)*; Antonio M Silva (Federal University of Maranhao, Brazil)*; Diego Augusto Santos Silva (Federal University of Santa Catarina, Brazil)*; Mary Simon (India Diabetes Research Foundation, India)*; Judith Simons (St Vincent's Hospital, Australia)*; Leon A Simons (University of New South Wales, Australia)*; Michael Sjostrom (Karolinska Institutet, Sweden)*; Jolanta Slowikowska-Hilczer (Medical University of Lodz, Poland)*; Przemyslaw Slusarczyk (International Institute of Molecular and Cell Biology, Poland)*; Liam Smeeth (London School of Hygiene & Tropical Medicine, UK)*; Margaret C Smith (University of Oxford, UK)*; Marieke B Snijder (Academic Medical Center of University of Amsterdam, The Netherlands)*; Hung-Kwan So (The

Chinese University of Hong Kong, China)*; Eugène Sobngwi (University of Yaoundé 1, Cameroon)*; Stefan Söderberg (Umeå University, Sweden)*; Moesijanti YE Soekatri (Health Polytechnics Institute, Indonesia)*; Vincenzo Solfrizzi (University of Bari, Italy)*; Emily Sonestedt (Lund University, Sweden)*; Yi Song (Peking University, China)*; Thorkild IA Sørensen (University of Copenhagen, Denmark)*; Maroje Soric (University of Zagreb, Croatia)*; Charles Sossa Jérome (Institut Régional de Santé Publique, West Africa)*; Aicha Soumare (University of Bordeaux, France)*; Jan A Staessen (University of Leuven, Belgium)*; Gregor Starc (University of Ljubljana, Slovenia)*; Maria G Stathopoulou (INSERM, France)*; Kaspar Staub (University of Zurich, Switzerland)*; Bill Stavreski (Heart Foundation, Australia)*; Jostein Steene-Johannessen (Norwegian School of Sport Sciences, Norway)*; Peter Stehle (Bonn University, Germany)*; Aryeh D Stein (Emory University, USA)*; George S Stergiou (Sotiria Hospital, Greece)*; Jochanan Stessman (Hadassah University Medical Center, Israel)*; Jutta Stieber (Helmholtz Zentrum München, Germany)*; Doris Stöckl (Helmholtz Zentrum München, Germany)*; Tanja Stocks (Lund University, Sweden)*; Jakub Stokwiszewski (National Institute of Public Health-National Institute of Hygiene, Poland)*; Gareth Stratton (Swansea University, UK)*; Karien Stronks (University of Amsterdam, The Netherlands)*; Maria Wany Strufaldi (Federal University of São Paulo, Brazil)*; Chien-An Sun (Fu Jen Catholic University, Taiwan)*; Johan Sundström (Uppsala University, Sweden)*; Yn-Tz Sung (The Chinese University of Hong Kong, China)*; Jordi Sunyer (ISGlobal Centre for Research in Environmental Epidemiology, Spain)*; Paibul Suriyawongpaisal (Mahidol University, Thailand)*; Boyd A Swinburn (The University of Auckland, New Zealand)*; Rody G Sy (University of the Philippines, Philippines)*; Lucjan Szponar (National Food and Nutrition Institute, Poland)*; E Shyong Tai (National University of Singapore, Singapore)*; Mari-Liis Tammesoo (University of Tartu, Estonia)*; Abdonas Tamosiunas (Lithuanian University of Health Sciences, Lithuania)*; Line Tang (Research Centre for Prevention and Health, Denmark)*; Xun Tang (Peking University Health Science Center, China)*; Frank Tanser (University of KwaZulu-Natal, South Africa)*; Yong Tao (Peking University, China)*; Mohammed Rasoul Tarawneh (Ministry of Health, Jordan)*; Jakob Tarp (University of Southern Denmark, Denmark)*; Carolina B Tarqui-Mamani (National Institute of Health, Peru)*; Anne Taylor (The University of Adelaide, Australia)*; Félicité Tchibindat (UNICEF, Cameroon)*; Holger Theobald (Karolinska Institutet, Sweden)*; Lutgarde Thijs (University of Leuven, Belgium)*; Betina H Thuesen (Research Centre for Prevention and Health, Denmark)*; Anne Tjonneland (Danish Cancer Society Research Centre, Denmark)*; Hanna K Tolonen (National Institute for Health and Welfare, Finland)*; Janne S Tolstrup (University of Southern Denmark, Denmark)*; Murat Topbas (Karadeniz Technical University, Turkey)*; Roman Topór-Madry (Jagiellonian University Medical College, Poland) *; Maties Torrent (IB-SALUT Area de Salut de Menorca, Spain)*; Stefania Toselli (University of Bologna, Italy)*; Pierre Traissac (Institut de Recherche pour le Développement, France)*; Antonia Trichopoulou (Hellenic Health Foundation, Greece)*; Dimitrios Trichopoulos (Harvard TH Chan School of Public Health, USA; deceased)*; Oanh TH Trinh (University of Pharmacy and Medicine of Ho Chi Minh City, Vietnam)*; Atul Trivedi (Government Medical College, India)*; Lechaba Tshepo (Sefako Makgatho Health Science University, South Africa)*; Marshall K Tulloch-Reid (The University of the West Indies, Jamaica)*; Tomi-Pekka Tuomainen (University of Eastern Finland, Finland)*; Jaakko Tuomilehto (Dasman Diabetes Institute, Kuwait)*; Maria L Turley (Ministry of Health, New Zealand)*; Per Tynelius (Karolinska Institutet, Sweden)*; Themistoklis Tzotzas (Hellenic Medical Association for Obesity, Greece)*; Christophe Tzourio (University of Bordeaux, France)*; Peter Ueda (Harvard TH Chan School of Public Health, USA)*; Flora AM Ukoli (Meharry Medical College, USA)*; Hanno Ulmer (Medical University of Innsbruck, Austria)*; Belgin Unal (Dokuz Eylul University, Turkey)*; Hannu MT Uusitalo (University of Tampere Tays Eye Center, Finland)*; Gonzalo Valdivia (Pontificia Universidad Católica de Chile, Chile)*; Susana Vale (University of Porto, Portugal)*; Damaskini Valvi (Harvard TH Chan School of Public Health, USA)*; Yvonne T van der Schouw (University Medical Center Utrecht, The Netherlands)*; Koen Van Herck (Ghent University, Belgium)*; Hoang Van Minh (Hanoi School of Public Health, Vietnam)*; Lenie van Rossem (University Medical Center Utrecht, The Netherlands)*; Irene GM van Valkengoed (Academic Medical Center of University of Amsterdam, The Netherlands) *; Dirk Vanderschueren (Katholieke Universiteit Leuven, Belgium)*; Diego Vanuzzo (Centro di Prevenzione Cardiovascolare Udine, Italy)*; Lars Vatten (Norwegian University of Science and Technology, Norway)*; Tomas Vega (Consejería de Sanidad Junta de Castilla y León, Spain)*; Gustavo Velasquez-Melendez (Universidade Federal de Minas Gerais, Brazil)*; Giovanni Veronesi (University of Insubria, Italy)*; WM Monique Verschuren (National Institute for Public Health and the Environment, The Netherlands)*; Roosmarijn Verstraeten (Institute of Tropical Medicine, Belgium)*; Cesar G Victora

(Universidade Federal de Pelotas, Brazil)*; Giovanni Viegi (Italian National Research Council, Italy)*; Lucie Viet (National Institute for Public Health and the Environment, The Netherlands)*; Eira Viikari-Juntura (Finnish Institute of Occupational Health, Finland)*; Paolo Vineis (Imperial College London, UK)*; Jesus Vioque (Universidad Miguel Hernandez, Spain)*; Jyrki K Virtanen (University of Eastern Finland, Finland)*; Sophie Visvikis-Siest (INSERM, France)*; Bharathi Viswanathan (Ministry of Health, Seychelles)*; Peter Vollenweider (Lausanne University Hospital, Switzerland)*; Sari Voutilainen (University of Eastern Finland, Finland)*; Ana Vrdoljak (UHC Zagreb, Croatia)*; Martine Vrijheid (ISGlobal Centre for Research in Environmental Epidemiology, Spain)*; Alisha N Wade (University of the Witwatersrand, South Africa)*; Aline Wagner (University of Strasbourg, France)*; Janette Walton (University College Cork, Ireland)*; Wan Nazaimoon Wan Mohamud (Institute for Medical Research, Malaysia)*; Ming-Dong Wang (Public Health Agency of Canada, Canada)*; Qian Wang (Xinjiang Medical University, China)*; Ya Xing Wang (Beijing Tongren Hospital, China)*; S Goya Wannamethee (University College London, UK)*; Nicholas Wareham (University of Cambridge, UK)*; Deepa Weerasekera (Ministry of Health, New Zealand)*; Peter H Whincup (St George's, University of London, UK) *; Kurt Widhalm (Medical University of Vienna, Austria)*; Indah S Widyahening (Universitas Indonesia, Indonesia)*; Andrzej Wiecek (Medical University of Silesia, Poland)*; Alet H Wijga (National Institute for Public Health and the Environment, The Netherlands)*; Rainford J Wilks (The University of the West Indies, Jamaica)*; Johann Willeit (Medical University Innsbruck, Austria)*; Tom Wilsgaard (UiT The Arctic University of Norway, Norway)*; Bogdan Wojtyniak (National Institute of Public Health-National Institute of Hygiene, Poland)*; Jyh Eiin Wong (Universiti Kebangsaan Malaysia, Malaysia)*; Tien Yin Wong (Duke-NUS Graduate Medical School, Singapore)*; Jean Woo (The Chinese University of Hong Kong, China)*; Mark Woodward (University of Sydney, Australia; University of Oxford, UK)*; Frederick C Wu (University of Manchester, UK)*; Jianfeng Wu (Shandong University of Traditional Chinese Medicine, China)*; Shou Ling Wu (Kailuan General Hospital, China)*; Haiquan Xu (Institute of Food and Nutrition Development of Ministry of Agriculture, China)*; Liang Xu (Capital Medical University, China)*; Uruwan Yamborisut (Mahidol University, Thailand)*; Weili Yan (Children's Hospital of Fudan University, China)*; Xiaoguang Yang (Chinese Center for Disease Control and Prevention, China)*; Nazan Yardim (Ministry of Health, Turkey)*; Xingwang Ye (University of Chinese Academy of Sciences, China)*; Panayiotis K Yiallouros (University of Cyprus, Cyprus)*; Akihiro Yoshihara (Niigata University, Japan)*; Qi Sheng You (Capital Medical University, China)*; Novie O Younger-Coleman (The University of the West Indies, Jamaica)*; Ahmad F Yusoff (Ministry of Health Malaysia, Malaysia)*; Ahmad A Zainuddin (Universiti Teknologi MARA, Malaysia)*; Sabina Zambon (University of Padova, Italy)*; Tomasz Zdrojewski (Medical University of Gdansk, Poland)*; Yi Zeng (Duke University, USA)*; Dong Zhao (Beijing Anzhen Hospital, Capital Medical University, China)*; Wenhua Zhao (Chinese Center for Disease Control and Prevention, China)*; Yingfeng Zheng (Singapore Eye Research Institute, Singapore)*; Maigeng Zhou (Chinese Center for Disease Control and Prevention, China)*; Dan Zhu (Inner Mongolia Medical University, China)*; Esther Zimmermann (Bispebjerg and Frederiksberg Hospitals, Denmark)*; Julio Zuñiga Cisneros (Gorgas Memorial Institute of Public Health, Panama).

## Additional information

### Funding

| Funder | Grant reference number | Author |
|---|---|---|
| Grand Challenges Canada | | Majid Ezzati |
| Wellcome Trust | 101506/Z/13/Z | Majid Ezzati |

The funders had no role in study design, data collection and interpretation, or the decision to submit the work for publication.

### Author contributions

NCD-RisC, collectively contributed to the research and manuscript. Members of the Country and Regional Data Group collected and reanalysed data, and checked pooled data for accuracy of information about their study and other studies in their country. MDC led data collection and JB led the statistical analysis and prepared results. Members of the Pooled Analysis and Writing Group collated

data, checked all data sources in consultation with the Country and Regional Data Group, analysed pooled data, and prepared results. ME designed the study, oversaw research, and wrote the first draft of the report with input from other members of Pooled Analysis and Writing Group. Members of Country and Regional Data Group commented on draft report.

## Additional files

**Supplementary files**

• Supplementary file 1. Regions used for the Bayesian hierarchical model such that information was shared among countries within each region, among regions in a super-region, and among super-regions in the world. Numbers in brackets show number of countries in each region or super-region.

• Supplementary file 2. Flowchart of secondary search for data sources.

• Supplementary file 3. Data sources used in the study, by country.

• Supplementary file 4. Results of model validation. The validation procedure is described in the main text.

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
