## [Decision Letter]

Thank you for submitting your work entitled "The height of the world – A century of trends in adult human height" for consideration by *eLife*. Your article has been reviewed by two peer reviewers, and the evaluation has been overseen by Eduardo Franco, as a Reviewing Editor, and Prabhat Jha, as the Senior Editor. M Dawn Teare, a Member of *eLife*'s Board of Reviewing Editors served as one of the reviewers and agreed to reveal her identity.

The reviewers have discussed the reviews with one another and the Reviewing Editor has drafted this decision to help you prepare a revised submission.

Summary:

This paper is a substantial and impressive report submitted on behalf of the NCD Risk Factor Collaboration. It represents a huge and extremely valuable new assemblage of data, including adult height measurements for around 15.4 million individuals born between 1896 and 1996 from 178 countries around the globe. Never before has such comprehensive data on stature been brought together, bearing on trends and differentials across the globe in health, nutrition, economy, and anthropometry. This collaborative group has published several papers now using this methodology, tackling a different health outcome in each paper. Here the health outcome is adult height and the team has systematically collected adult height measurements from 1450 studies from 178 countries for adults born between 1896 and 1996 and used a hierarchical Bayesian model to analyse trends over 100 years. Adult height lends itself nicely to this sort of analysis as the assumption is that height is pretty constant after 18, whereas weight (and hence BMI) is a much less stable measurement.

Essential revisions:

Structure and Organization:

The paper is difficult to read and some investment in visualisation tools would greatly enhance its value. The maps after the References section are very nice and easy to fit into the article in a pdf form. However, the country by gender plots need to be made a bit more accessible as these are the more interesting results summarizing the trends.

Please revise the Abstract and Introduction with due attention to providing factual material only. As they stand, the findings listed in the Abstract and Introduction would hardly justify publication. They are tidbits, engaging the curiosity of readers and showing off the scope of the assembled data, but not settling open questions of theoretical interest. Everyone knows that nutritional status across the world has not converged to some common level. Finding “no indication of convergence across countries” in mean adult heights is hardly news.

The paper has a number of strengths that do not come across in the Abstract. They have systematically searched for sampled measured data rather than self-reports; they have collected a large amount of data on women and have data from 178 countries. This means that the work is a substantial step up from other studies of trends in height. I think the paper is too short. Please explain the BMI analysis referred to in Figure 6.

The main text and figures are valuable. The 165 pages of Supplementary Information, in contrast, do not belong in the publication. The lists of NCD Risk Factor Collaborators and the long table of data sources belong on a Project Website with hyperlink pointers in the article, or perhaps as a separate appendix hosted in the journal. Some details of the validation study might reasonably belong an appendix, but the validation study as it stands is not entirely convincing. The uncertainties of importance relate to the out-of-sample-range extrapolations to timeframes and countries without datasets, whereas the cross-validation mainly measures success at within-sample-range interpolations within sets of times and cases where relevant datasets are available.

Data Analysis:

What is the specification of the Bayesian model in use here for filling in missing data and extrapolating back into the past and outward to nations with limited sets of direct measurements? The paper directs readers to Danaei et al., 2011 and Finucane et al., 2014 for details of the model, but the Bayesian models in those references pertain to systolic blood pressure and to health status, not to heights. Heights pose many different issues, particularly when only 70 of the 178 countries have data for cohorts born before 1920 and 22 of 200 countries for which estimates are generated have no data at all. Presumably, the model here incorporates features needed for application to heights, but nothing is spelled out. Toward the end of this review is a list of some of the model features that would seem important to describe.

In what form and under what arrangements are these data to be made available to the wider community of researchers? Is the creation of a data resource for heights along the lines of the Human Mortality Database and the Human Fertility Database underway? This question arises not only with regard to compliance with data-sharing requirements of *eLife* and other top journals, but also with regard to the wide range of scientific questions that could be addressed with these data. What is already treated in this paper hardly scratches the surface.

Please provide details on the Bayesian model regarding the following:

A) growth curves by age;

B) the “linear and non-linear” trends in mean age over time;

C) non-normality at younger and older ages;

D) variability in standard deviations and its relationship to the homogeneity or heterogeneity of each measured population;

E) smoothing (B-splines?);

F) covariance structures within region by age and time;

G) sample information with regard to measurement scales in centimeters or inches, with or without shoes (or unknown), degree of rounding, etc.

---

## [Author Response]

Essential revisions:

Structure and Organization:

The paper is difficult to read and some investment in visualisation tools would greatly enhance its value. The maps after the References section are very nice and easy to fit into the article in a pdf form. However, the country by gender plots need to be made a bit more accessible as these are the more interesting results summarizing the trends.

We take this comment to refer to old Figure 3 (new Figure 4 and Figure 5), the long “ladder” plot of all countries. As the reviewer has correctly pointed out, with 200 countries, this figure would be best suited to a dynamic visualisation. We have developed the needed technology (see http://www.ncdrisc.org/ranking-bmi.html for what it looks likes for BMI) and will release similar graphs for height upon the paper’s publication. If *eLife* system allows, the current static figure can be replaced with an embedded dynamic one on the Journal’s website while the static one appears in the PDF. We would be happy to finalise the specifics with the Editors and the production team.

Please revise the Abstract and Introduction with due attention to providing factual material only. As they stand, the findings listed in the Abstract and Introduction would hardly justify publication. They are tidbits, engaging the curiosity of readers and showing off the scope of the assembled data, but not settling open questions of theoretical interest. Everyone knows that nutritional status across the world has not converged to some common level. Finding “no indication of convergence across countries” in mean adult heights is hardly news.

We have done as suggested for Abstract. The Introduction contains only a concise summary of current literature in the field, and the contribution of the paper; to the best of our ability, it is entirely factual.

The paper has a number of strengths that do not come across in the Abstract. They have systematically searched for sampled measured data rather than self-reports; they have collected a large amount of data on women and have data from 178 countries. This means that the work is a substantial step up from other studies of trends in height. I think the paper is too short. Please explain the BMI analysis referred to in Figure 6.

We understand that the Abstract is restricted to ~150 words. With this constraint in mind, we have stated that the data used in the paper were from “reanalysis” of “population-based studies with measurement of height” and have stated the number of studies and participants.

The paper that presents the BMI analysis is now published and has been cited (NCD Risk Factor Collaboration, 2016), together with a concise statement on its scope. We would be happy to provide more information in this paper if it is deemed informative to repeat the materials here.

The main text and figures are valuable. The 165 pages of Supplementary Information, in contrast, do not belong in the publication. The lists of NCD Risk Factor Collaborators and the long table of data sources belong on a Project Website with hyperlink pointers in the article, or perhaps as a separate appendix hosted in the journal. Some details of the validation study might reasonably belong an appendix, but the validation study as it stands is not entirely convincing. The uncertainties of importance relate to the out-of-sample-range extrapolations to timeframes and countries without datasets, whereas the cross-validation mainly measures success at within-sample-range interpolations within sets of times and cases where relevant datasets are available.

We request to keep the list of data sources because it is increasingly the norm in presenting global health estimates to state the data sources used in the analysis (soon to become a part of reporting guidelines). We have removed the country-specific graphs as suggested, and will show these on NCD-RisC website upon the paper’s publication.

Our data use agreement with our collaborators requires listing all of the authors, which has been done even in print journals (see for example http://www.thelancet.com/journals/lancet/article/PIIS0140-6736(16)30054-X/fulltext and http://www.thelancet.com/journals/lancet/article/PIIS0140-6736(16)00618-8/fulltext; see http://www.ncbi.nlm.nih.gov/pubmed/26109024 for how this appears in PubMed). Therefore, we request to be allowed to include the full list of authors especially given that *eLife* is published online.

The validation analysis is entirely out of sample validation. We removed data for specific countries (i.e. they are no longer in the sample of countries with data) and tested how well the model predicts the known-but-withheld/removed data. We have attempted to clarify this procedure in the revised paper (subsection “Author contributions”).

Data Analysis:

What is the specification of the Bayesian model in use here for filling in missing data and extrapolating back into the past and outward to nations with limited sets of direct measurements? The paper directs readers to Danaei et al., 2011 and Finucane et al., 2014 for details of the model, but the Bayesian models in those references pertain to systolic blood pressure and to health status, not to heights. Heights pose many different issues, particularly when only 70 of the 178 countries have data for cohorts born before 1920 and 22 of 200 countries for which estimates are generated have no data at all. Presumably, the model here incorporates features needed for application to heights, but nothing is spelled out. Toward the end of this review is a list of some of the model features that would seem important to describe.

One of the two papers cited is a methodological paper that lays out the Bayesian model used here; the appendix of the other has full model specification even if the paper’s substance involves its application to blood pressure. Nonetheless, as suggested below, we have added additional details regarding the model (subsection “Statistical methods”, fourth paragraph).

In what form and under what arrangements are these data to be made available to the wider community of researchers? Is the creation of a data resource for heights along the lines of the Human Mortality Database and the Human Fertility Database underway? This question arises not only with regard to compliance with data-sharing requirements of eLife and other top journals, but also with regard to the wide range of scientific questions that could be addressed with these data. What is already treated in this paper hardly scratches the surface.

NCD-RisC is a data pooling analysis that uses secondary data. Some of our data are from public sources and we would be happy to point others to the relevant sites for these sources, or provide the data. Other sources are provided either by specific scientists or national health agencies. For these, we will be happy to provide contact information of the data provider for requests to be made.

Please provide details on the Bayesian model regarding the following:

A) growth curves by age;

We have specified the age component of the model (subsection “Statistical methods”, fourth paragraph), noting that we model population mean height over age from age 18, so it is not growth at the individual level.

B) the “linear and non-linear” trends in mean age over time;

Done (subsection “Statistical methods”, fourth paragraph).

C) non-normality at younger and older ages;

We do not model individual height, for which non-normality may be an issue. Rather our analysis models mean height of the population, and its distribution across countries and health surveys. We rely on standard central limit theorem as justification for treating *mean height* as normally distributed across countries/surveys (Finucane et al., 2014) with error around the true population mean, which is the quantity of interest. We also considered (and have used elsewhere (Stevens et al, 2012; Finucane et al., 2015) a t_4_ distribution which, by having heavier tails, allows for outlier studies. The results of the current analysis were not sensitive to choice, confirming that the normal prior appropriately described the distribution of mean height.

D) variability in standard deviations and its relationship to the homogeneity or heterogeneity of each measured population;

As mentioned above, the analysis and modelling are of the mean height, so the only relevant standard deviation is the standard error of the sample means. Standard errors were computed together with sample means when NCD-RisC members re-analysed each data source. The standard deviations of each data source study are reflected in the standard errors used in specifying the distribution of the sample means.

E) smoothing (B-splines?);

The smoothing of time trends is done using a 2^nd^ order conditional auto-regressive model (also known as random walk), specified in the revised paper with appropriate citation (subsection “Statistical methods”, second paragraph).

F) covariance structures within region by age and time;

The covariance between different birth cohorts (i.e., the time scale in our model) is induced by the conditional auto-regressive structure. Formally, the auto-regressive model induces a particular precision (inverse covariance) structure for cohorts within a country and the induced covariance is therefore the inverse of that. The linear and non-linear components of this auto-regressive structure, as well as its intercept, are modelled hierarchically so the effects for each country borrow from the region to the extent that the data suggest that countries within a region have similar levels and trends across cohort. This induces covariance between countries within a region, even if not modelled explicitly. Such hierarchical structure is a standard strategy for accounting for dependence in statistical modelling. With regard to age, as stated in the revised paper, population mean adult height declined by only a small amount as the birth cohort ages (subsection “Statistical methods”, second paragraph). Nonetheless, the use of an age model introduces dependence in height over age.

G) sample information with regard to measurement scales in centimeters or inches, with or without shoes (or unknown), degree of rounding, etc.

A few data sources were in inches or meters and were converted to centimetres, which is of course an entirely deterministic calculation. To our knowledge, self-report is the single most important source of bias in adult height data. A major strength of our paper is the exclusive use of measured data and the exclusion of self-reported height. Removal of shoes is a part of the standard protocol of health and nutrition surveys (Madden et al, 2016). Our sources are high-quality health/nutrition surveys and epidemiological studies, and we expect removing shoes to be a part of their protocol. We have nonetheless stated measurement error as a potential limitation of population-based data (Discussion, sixth paragraph).

References

Finucane MM, Paciorek CJ, Stevens GA, Ezzati M. Semiparametric Bayesian density estimation with disparate data sources: a meta-analysis of global childhood undernutrition. J Am Stat Assoc 2015; 110(511): 889-901.

Madden AM, Smith S. Body composition and morphological assessment of nutritional status in adults: a review of anthropometric variables. J Hum Nutr Diet 2016; 29(1): 7-25.